# Topologically controlled synthesis of active colloidal bipeds

Jonas Elschner [1], Farzaneh Farrokhzad [1], Piotr Kuświk [2], Maciej Urbaniak [2], Feliks Stobiecki[2], Sapida Akhundzada [3], Arno Ehresmann [3], Daniel de las Heras [4] & Thomas M. Fischer [1] ✉

Topological growth control allows to produce a narrow distribution of out-grown colloidal rods with defined and adjustable length. We use an external magnetic field to assemble paramagnetic colloidal spheres into colloidal rods of a chosen length. The rods reside above a metamorphic hexagonal magnetic pattern. The periodic repetition of specific loops of the orientation of an applied external field renders paramagnetic colloidal particles and their assemblies into active bipeds that walk on the pattern. The metamorphic patterns allow the robust and controlled polymerization of single colloids to bipeds of a desired length. The colloids are exposed to this fixed external control loop that causes multiple simultaneous responses: Small bipeds and single colloidal particles interpret the external magnetic loop as an order to walk toward the active zone, where they assemble and polymerize. Outgrown bipeds interpret the same loop as an order to walk away from the active zone. The topological transition occurs solely for the growing biped and nothing is changed in the environment nor in the magnetic control loop. As in many biological systems the decision of a biped that reached its outgrown length to walk away from the reaction site is made internally, not externally.

The length of a one-dimensional passive assembly of periodically repeated units is the most important structural quantity of the assembly that determines its physical equilibrium and non-equilibrium properties. Growth control of the assembly in order to achieve the *desired length* therefore is a scientific question of immediate technological relevance in molecular[1] and supramolecular[2] polymer chemistry, for the growth of nanotubes[3] and nanowires[4] as well as in biology for the control of the length of DNA[5], for the growth of flagella[6] or for the switching from vegetative plant growth toward reproductive growth[7]. The mechanism of length control in these different systems can be via an external feedback[1,3,7], where the environment (an inhibitor, a mask, or the daylight duration via a systemic signal, called florigen) tells the growing system that the end (of polymerization time, of the mask, or of vegetative growth) has been reached. An alternative length control occurs via a balance between the growth and the

decomposition kinetics[1,2,6]. A third interesting mechanism is the use of coprime repetitions of complementary single strands of DNA[5] that leads to a stop of the growth of the forming double-stranded DNA once the single strands have bound to the double-strand product length of units of the two coprime numbers. These mechanisms for external length control share the passive role of the assembly in determining when its length is long enough. An active entity, in contrast, must not be told externally but should be able to make an internal decision when its proper length has been reached.

Here, we use a topological and, therefore, robust transition of the internal interpretation of an external magnetic control loop for the control of the length of walking magnetic multi-colloidal bipeds. The bipeds actively walk on a metamorphic magnetic pattern. A metamorphic pattern is a quasi-periodic pattern with unit cells that continuously change as one moves along the pattern. Small bipeds and

[1]Experimentalphysik X, Physikalisches Institut, Universität Bayreuth, Bayreuth, Germany. [2]Institute of Molecular Physics, Polish Academy of Sciences, Poznań, Poland. [3]Institute of Physics and Center for Interdisciplinary Nanostructure Science and Technology (CINSaT), Universität Kassel, Kassel, Germany. [4]Theoretische Physik II, Physikalisches Institut, Universität Bayreuth, Bayreuth, Germany. ✉e-mail: Thomas.Fischer@uni-bayreuth.de

single colloidal particles interpret the external magnetic loop as an order to walk from a random initial position toward the active zone, where they assemble and polymerize, while large enough bipeds interpret the same loop as an order to walk away from the active zone. The topological transition occurs solely for the growing biped and nothing is changed in the environment nor in the magnetic control loop. Hence the decision to walk away from the reaction site is made internally, not externally.

## Results

### Experimental setup

Paramagnetic colloidal particles (diameter $d = 2.8\,\mu m$) immersed in water are placed on top of a two-dimensional magnetic pattern. The pattern is a metamorphic hexagonal lattice of alternating regions with positive and negative magnetization relative to the direction normal to the pattern, see Fig. 1a. A uniform time-dependent external field of constant magnitude is superimposed to the non-uniform time-independent magnetic field generated by the pattern. The external field induces strong dipolar interactions between the colloidal particles which respond by self-assembling into bipeds of $n^*$ particles, provided sufficient single colloidal particles are available[8].

In Fig. 1b we show the orientation of the external field $\boldsymbol{H}_{ext}$ (black) of fixed magnitude that adiabatically varies along a closed loop in control space $\mathcal{C}$ – a sphere of radius $H_{ext}$. Despite the field returning to its initial direction, single colloidal particles can be topologically transported by one unit cell after completion of one loop[9–11]. Their

transport is topologically protected but passive in the sense that they are carried along in the moving local minimum of the potential. There exist an entire family of periodic hexagonal patterns. Each pattern in this family is characterized by a specific value of a symmetry phase[12]. The transport in an unmorphed periodic hexagonal pattern occurs provided that the loop winds around specific orientations of the external field called bifurcation points that depend on the symmetry phase of the pattern. In our metamorphic pattern, the symmetry phase varies with the location on the pattern, and this allows us to use well-designed loops where single colloids are transported in different directions in different regions of the metamorphic pattern[13]. Colloidal bipeds formed by several particles can also be transported[8,14]. The biped aligns with the external field since dipolar interactions are stronger than the buoyancy. Hence, if the external field is not parallel to the pattern, one end of the biped (a foot) remains on the ground while the other one is lifted. As a result the bipeds actively walk by alternatingly grounding one of their feet. In addition, as for the single colloids, the grounded foot passively slides above the pattern[8,14,15].

To transport the bipeds, the loop also needs to wind around special orientations of the external field that depend on the length of the bipeds and on the pattern's symmetry phase. Bipeds of different lengths can fall into different topological classes such that their displacement upon completion of one loop can be different in both magnitude and direction. A sketch of the process is shown in Fig. 1. As it is the case for single colloids, the biped motion is topologically protected and hence, robust against perturbations.

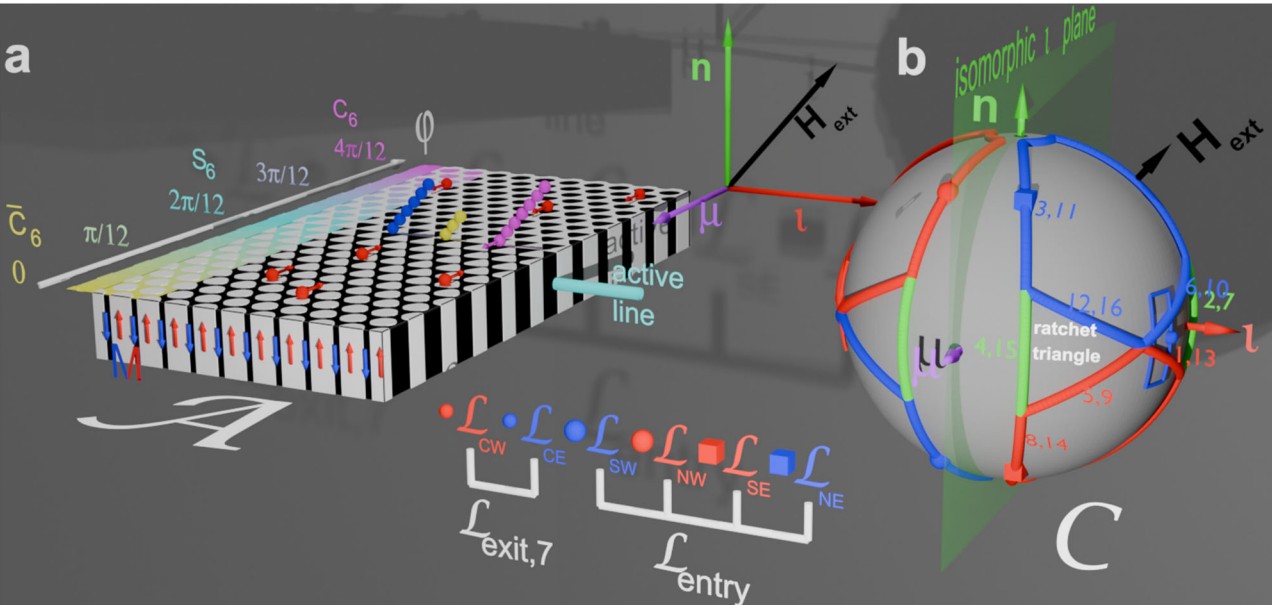

**Fig. 1 | Schematic of the setup. a** A metamorphic hexagonal magnetic pattern consists of domains of positive (white) and negative (black or colored) magnetization parallel to the normal vector of the pattern $\mathbf{n}$. Along the metamorphic $\boldsymbol{\mu}$-direction the pattern morphs from a hexagonal $C_6$ pattern with down magnetized circular bubble domains via an improper six-fold symmetric $S_6$-pattern to a hexagonal $\bar{C}_6$ bubble pattern with up magnetized bubbles. Along the isomorphic $\iota$-direction the pattern remains periodic. The vectors $\mathbf{n}, \boldsymbol{\mu}$ and $\iota$ are not drawn to scale. Colloidal particles (red) are placed on top of the pattern immersed in water. Due to the presence of a strong homogeneous external field (black), some colloids self-assemble into bipeds of juvenile (yellow or blue) and outgrown (magenta) length. The 2$d$-space where the colloids move is called the action space $\mathcal{A}$. **b** Our control space $\mathcal{C}$ is a sphere that represents all possible directions of the external field $\mathbf{H}_{ext}$ (black). The direction of the external field varies in time, performing a loop $\mathcal{L} = \mathcal{L}_{entry} * \mathcal{L}_{exit}$ (a closed path) that is a concatenation of a (thicker) entry loop $\mathcal{L}_{entry}$ and a (thinner) exit loop $\mathcal{L}_{exit}$. Both loops consist of sub-loops revolving in the mathematical positive (blue) and negative (red) sense. The sub-loops consist of segments (open paths) that connect different corners of the sub-loop where the

direction of the loop changes. The time sequence of consecutive segments of the eastern part of the entry loop is indicated by numbers shown with the loop segments. The black arrow tip on the loop corresponds to the orientation of the external field depicted in (**a**). The loops are eigen loops of the $\sigma_\iota$ mirror operation and, therefore, suppress any net motion into the isomorphic $\iota$-direction. At any time during the modulation the orientation of the bipeds in (**a**) is parallel to the orientation of the external magnetic field $\mathbf{H}_{ext}$. Bipeds in (**a**) and the orientation of $\mathbf{H}_{ext}$ in (**b**) are shown at the same time. We find entry loops that transport single colloids and small bipeds toward the active line (cyan) on the $S_6$ isomorphic line of the pattern. The bipeds grow in this region via dipolar attraction due to the high particle density. Once they have reached the outgrown length (see magenta biped) encoded by our control exit loop, the bipeds topologically change and reinterpret the same control loop as a command to walk away from the $S_6$ region into the metamorphic $\boldsymbol{\mu}$-direction. The loop $\mathcal{L}_{CW}$ is not visible in the figure but is a mirror image of the loop $\mathcal{L}_{CE}$ mirrored at the isomorphic $\iota$-plane. We show it in Supplementary Video 1, where we record the scheme from different perspectives.

In this work we show that one can apply external field modulations that act in synergy with the metamorphic pattern to synthesize bipeds of a chosen length. The choice of metamorphic pattern and external field modulation must be such that single colloidal particles and bipeds of different length all fall into the wanted topological transport class at each of the different locations.

## Metamorphic pattern

We choose a pattern with a thin film of magnetization

$$\mathbf{M} = M\delta(qz)\mathbf{n}\,\text{sign}\left(\frac{1}{2}\cos(3\varphi) + \sum_{i=0}^{2}\cos(\mathbf{q}_i\cdot\mathbf{r}_\mathcal{A} - \varphi)\right) \qquad (1)$$

where $\mathbf{r}_\mathcal{A}$ is the two-dimensional vector in action space $\mathcal{A}$, which is the plane parallel to the pattern where the colloids move. The modulus of the saturation magnetization is denoted by $M$, $\mathbf{n}$ is the vector normal to the film, the $z$ coordinate runs in the direction of $\mathbf{n}$ and $\varphi$ is the symmetry phase. If we fix the symmetry phase ($\nabla_\mathcal{A}\varphi = 0$) the pattern is a periodic hexagonal pattern of period $a$ (in our experiments $a = 7\,\mu m$). The vectors $\mathbf{q}_i = \frac{2\pi}{a\sin\pi/3}\mathbf{R}_{2\pi/3}^i\cdot\mathbf{e}_x$, $(i = 0, 1, 2)$, are three coplanar reciprocal unit vectors, and $\mathbf{R}_{2\pi/3}$ is a rotation matrix around the normal vector $\mathbf{n}$ by $2\pi/3$. The modulus $q$ of all three in-plane reciprocal unit vectors is the same.

If we render the symmetry phase $\varphi(\mathbf{r}_\mathcal{A})$ a function of action space this breaks the discrete translational symmetry. In our case we use a linear variation of the symmetry phase

$$\varphi = \frac{\pi}{3} + \boldsymbol{\mu}\cdot\mathbf{r}_\mathcal{A} \qquad (2)$$

with a small and fixed morphing reciprocal vector $\boldsymbol{\mu} = -\mathbf{q}_0/240$ of modulus $\mu \ll q$. Due to the small value of the morphing reciprocal vector, the lattice adiabatically morphs from one locally periodic pattern shape to the next as we move along the metamorphic $\boldsymbol{\mu}$-direction, see Fig. 1a. At the origin, the local pattern resembles a $C_6$ symmetric pattern (see magenta region in Fig. 1a, $\varphi \approx 4\pi/12$) with hexagonally arranged downward magnetized domains. When we move in the metamorphic direction, the local pattern morphs into an $S_6$-symmetric pattern at $\varphi \approx 2\pi/12$ with triangular upward-magnetized domains neighbored by oppositely oriented (cyan) down-magnetized domains. When the symmetry phase decreases to $\varphi \approx 0$, the local pattern resumes a local $\bar{C}_6$-symmetry with hexagonally arranged (white) upward magnetized domains in an extended downward (yellow) surrounding. The bar above the $\bar{C}_6$-symmetry indicates that the pattern is inverted with respect to the $C_6$-case, i.e., $\varphi \approx 4\pi/12$. At a symmetry phase of $\varphi \approx 6\pi/12$ (not shown), the symmetry is $\bar{S}_6$ with inverted triangles compared to the $2\pi/12$-case. Iso-phase lines run along the isomorphic vector $\boldsymbol{\iota}$, the periodic direction of the pattern with translation symmetry modulo the primitive unit vector $\mathbf{a}_1$.

In analogy to the active site of an enzyme, we call the iso-phase line with locally $S_6$-symmetric pattern at $\varphi = 2\pi/12$ the active line (cyan in Fig. 1a). The active line serves as a polymerization line for monomeric single colloidal particles into polymeric bipeds being rods of several colloidal spheres. The bond between the monomers of the polymerized bipeds is a physical dipolar interaction bond and not a chemical bond. The concentration of colloids outside the active region must be small enough to prevent polymerization reactions outside the active zone; polymerization addition reactions shall only occur in the active zone. There, our modulation loop will ensure that the reaction proceeds toward the wanted outgrown biped length and not toward unwanted longer biped lengths. Juvenile bipeds having lengths below the outgrown length shall not leave the active zone before they are outgrown.

## Modulation loop

In Fig. 1b we show the control space $\mathcal{C}$ and the modulation loop controlling the dynamic assembly of single colloidal particles toward an outgrown biped of length $b_{n^*} = b_7 = 7d$, where $d$ is the diameter of a single colloidal particle. The loop $\mathcal{L} = \mathcal{L}_{entry} * \mathcal{L}_{exit}$ is a concatenation of two loops: a (thick) entry loop $\mathcal{L}_{entry}$ and a (thin) exit loop $\mathcal{L}_{exit}$. The entry loop consists of the north-western subloop $\mathcal{L}_{NW}$, the south-western subloop $\mathcal{L}_{SW}$, the north-eastern subloop $\mathcal{L}_{NE}$, and the south-eastern subloop $\mathcal{L}_{SE}$. They wind in the mathematically positive (blue) and the mathematically negative (red) sense. Furthermore, we make use of (green) detour segments that we move along in both directions. The sequence of consecutive segments of the eastern part of the entry loop follows the numbers attached in Fig. 1b to each segment of the loop. The entry loop is an eigen loop $\mathcal{L}_{entry} = \sigma_n(\mathcal{L}_{entry}) = \sigma_t(\mathcal{L}_{entry})$ to the mirror operations reflecting at the equatorial $\mathbf{n}$- and the iso-morphic $\boldsymbol{\iota}$-plane, and the sequence of segments of the western part of $\mathcal{L}_{entry}$ can be inferred from the $\sigma_t$ mirror symmetry.

The green loop segments of the concatenated eastern loop $\mathcal{L}_{NE} * \mathcal{L}_{SE}$ correspond to the convex envelope of the eastern loop. The entry loop sequence in Fig. 1b is such that the north-eastern loop $\mathcal{L}_{NE}$ and the south-eastern loop are alternately extended each by one or the other of the (red, blue, green) triangles that, for reasons that will become clear later we call the ratchet triangles.

The exit loop $\mathcal{L}_{exit}$ consists of the two (thin) subloops $\mathcal{L}_{CW}$ and $\mathcal{L}_{CE}$. The exit loop $\mathcal{L}_{exit} = \sigma_t(\mathcal{L}_{exit})$ shares the mirror symmetry at the $\boldsymbol{\iota}$-plane with the entry loop (see the loop in the Supplementary Video 1). However, in contrast to the entry loop, its reflection at the $\mathbf{n}$-plane yields the inverse (time-reversed) exit loop $\mathcal{L}_{exit}^{-1} = \sigma_n(\mathcal{L}_{exit})$. We use the exit loop to move bipeds of the outgrown length $b_{n^*}$ out of the active zone. In the experiments we use six different exit loops $\mathcal{L}_{exit,n^*}$, with $n^* = 2, 3, 4, 5, 6, 7$, depending on the outgrown biped length $b_{n^*}$ that we intend to synthesize. In Fig. 1b we show the exit loop $\mathcal{L}_{exit,7}$ that is designed to produce bipeds of length $b_7$. Like the exit loop $\mathcal{L}_{exit,7}$ four of the other exit loops $\mathcal{L}_{exit,n^*}$, with $n^* = 2, 3, 5, 6$, also circle two $\boldsymbol{\iota}$-symmetric centers each. Their centers are, however, at positions on the equator different from that of $\mathcal{L}_{exit,7}$ and their azimuthal width is broader, the smaller the desired length $b_{n^*}$ to be synthesized.

## Synthesis of $b_3$-bipeds

We can design loops to grow bipeds of arbitrary length (for practical size limits see the discussion section). As an illustration, we have depicted four reflection microscopy snapshots in Fig. 2a, showing the colloidal particles on the metamorphic pattern subject to the loop $\mathcal{L} = \mathcal{L}_{entry} * \mathcal{L}_{exit,3}$ of period $T$ at different times. Six red single colloidal particles, one of them not yet in the field of view, are transported consecutively toward the $S_6$ symmetric region as a result of the loop. Two of these particles hop back and forth until a polymer addition reaction assembles them into an orange juvenile $b_2$-biped (time $t = 1.2\,T$). The juvenile biped waits in the active zone until a third red single colloidal particle is brought into the reaction zone. One further addition reaction leads to a yellow outgrown $b_{n^*} = b_3$ biped at $t = 2\,T$. The outgrown biped of length $b_3$ is transported into the metamorphic $\boldsymbol{\mu}$-direction as soon as it reaches its outgrown length. During the entire process (from $t = 0$ to $t = 3.5\,T$) two further single colloidal particles are approaching the active zone and will assemble to a second biped (shown in Supplementary Video 3 but not shown in Fig. 2a). In Fig. 2b and c we plot the tracked metamorphic and isomorphic coordinate of the single colloids respectively the metamorphic and isomorphic coordinate of the northern foot of the bipeds as a function of time. While the isomorphic coordinates in Fig. 2c are purely periodic with the period of the loop (indicated by the shading of the background), the metamorphic coordinates of all single colloids in Fig. 2b are aperiodic and progressively approach the $S_6$-active line from either side. Once in the active zone, the motion of the colloids switches to

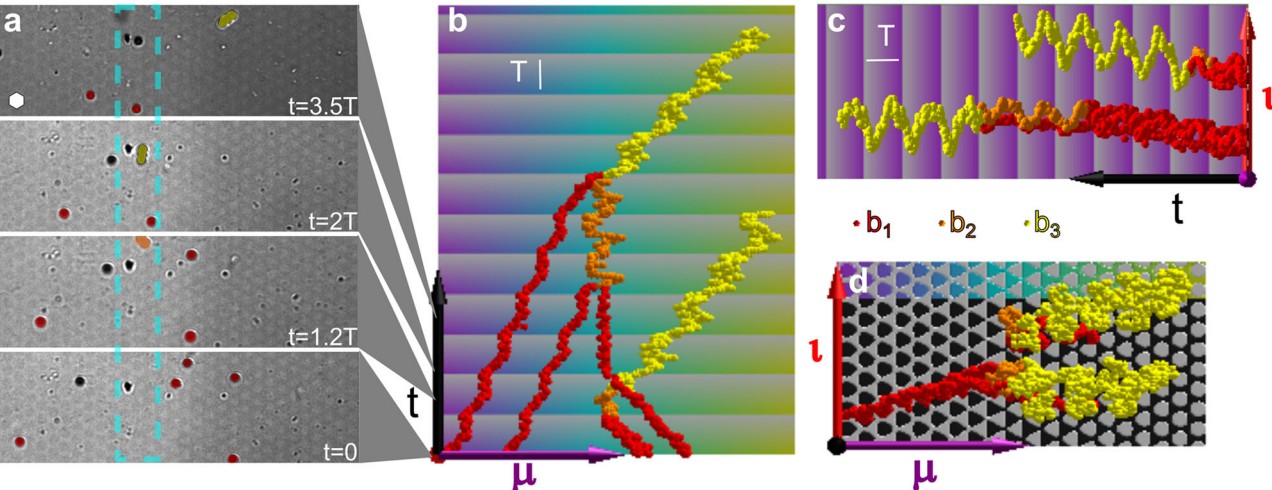

**Fig. 2 | $b_3$-synthesis. a** Four reflection microscopy images of the colloidal particles above the pattern subject to the loop $\mathcal{L} = \mathcal{L}_{entry}*\mathcal{L}_{exit,3}$ of period $T$. The hexagonal unit cell of the pattern (white hexagon) has lattice constant $a = 7\,\mu m$. The images show the synthesis and transport of the first (yellow) $b_3$ biped as well as the collection of single bipeds (red) in preparation of the second $b_3$ biped. The active zone is marked in cyan. **b** Trajectories of single colloidal particles and bipeds (colored according to their length $b_n$) as a function of the time $t$ and the metamorphic coordinate $\mu$. The background is colored according to the symmetry phase in the metamorphic direction, and the brightness of the background is periodic with the period $T$ of the modulation loop. **c** Trajectories of single colloidal particles and bipeds as a function of the time $t$ and the isomorphic coordinate $\iota$. **d** Trajectories of single colloidal particles and bipeds as a function of the metamorphic coordinate and the isomorphic coordinate. The background shows the magnetization pattern of the magnetic thin film. Supplementary Video 3 of the synthesis of two $b_3$-bipeds is provided with the supplementary information.

periodic until the motion is disrupted by a polymerization addition event. Finally, after the outgrown length $b_3$ is reached, the $b_3$-bipeds walk away into the positive metamorphic $\mu$-direction until they leave the field of view of the microscope. The trajectories are depicted above the pattern in Fig. 2d as a function of the isomorphic and metamorphic coordinates. We can see that the active line of polymerization is indeed at the $S_6$-symmetric line of the pattern. Supplementary Video 3 of the synthesis of two $b_3$-bipeds is provided with the supplementary information.

## Synthesis of $b_7$-bipeds

In Fig. 3a and b, we show the two reflection microscopy snapshots of the colloidal particles on the metamorphic pattern subject to the loop $\mathcal{L} = \mathcal{L}_{entry}*\mathcal{L}_{exit,7}$ at times $t = 0$ and $t = 7\,T$. In Fig. 3c and d we plot the tracked metamorphic and isomorphic coordinates of the single colloids respectively the metamorphic and isomorphic coordinates of the northern foot of the bipeds as a function of time. Figure 3e shows the trajectories above the pattern for the two in-plane directions. The motion is similar to that in Fig. 2. The only difference is that juvenile bipeds $b_n < b_7$ now remain in the active zone until they have reached their newly set outgrown length of $b_{n^*} = b_7$ before they leave this region and walk away into the metamorphic $\mu$-direction. Figure 3 shows one unsuccessful and two successful synthesis attempts. In the unsuccessful attempt, a $b_6$-biped (blue) is synthesized in the active zone. The lonely $b_6$-biped, however, is unsuccessful in attracting a final colloidal particle to reach the outgrown (magenta) $b_{n^*} = b_7$ length that would allow it to exit the active zone. In contrast to its successfully outgrown $b_7$-bipeds in the neighborhood, the $b_6$-biped continues its lonely back-and-forth walk inside the active zone without ever being allowed to leave. The topological constraint robustly keeps the $b_6$-biped inside the active zone. Supplementary Video 7 of the synthesis of one $b_6$-biped and two $b_7$-bipeds is provided with the supplementary information.

## Synthesis of $b_2$, $b_4$, $b_5$, and $b_6$-bipeds

We have also used four further exit loops $\mathcal{L}_{exit,n^*}$ to synthesize outgrown bipeds of length $b_{n^*}$ for $n^* = 2, 4, 5, 6$. Four videoclips: Supplementary Video 2, Supplementary Video 4, Supplementary Video 5, and Supplementary Video 6 are provided with the supplementary information. Details of the loops $\mathcal{L}_{exit,n^*}$ are provided in the methods section. For $n^* = 2$ one outgrown $b_2$-biped already occupies the active zone, a second is synthesized there, both are transported away, and five single colloidal particles remain in the active zone; for $n^* = 4$ two outgrown $b_4$-bipeds are synthesized in the active zone, transported away, and two single colloidal particles remain in the active zone; for $n^* = 5$ three outgrown $b_5$-bipeds are synthesized in the active zone, transported away, and one juvenile $b_3$-biped remains in the active zone; and for $n^* = 6$ one outgrown $b_6$-biped is synthesized in the active zone, transported away, and three single colloidal particles remain in the active zone. For any biped lengths, there are, in principle exit loops leaving juvenile bipeds in the active zone and transporting outgrown bipeds out of the active zone. The robustness of such loops, however, decreases with the biped length and whenever the biped length becomes commensurate with the lattice. Also, the period $T$ leading to adiabatic behavior increases with the desired outgrown biped length.

## Theoretical results

We have experimentally shown that the application of the loop $\mathcal{L}$ in control space to a collection of single colloidal particles on the metamorphic pattern successfully and without external interference collects single colloidal particles at the active line, adds them to juvenile bipeds and finally transports them into the metamorphic direction once they have reached their outgrown length $b_{n^*}$.

The rest of this work theoretically proves the topological nature of the programmed outgrown synthesis and explains the topology behind the design of the loops as well as the topological response of the colloidal assemblies to those loops in the different regions of action space.

## Single colloidal particle response

In Fig. 4 we replot the magnetic pattern in action space $\mathcal{A}$ together with copies of the control spaces of single colloidal particles for the locally periodic regions for seven symmetry phases. The single colloidal

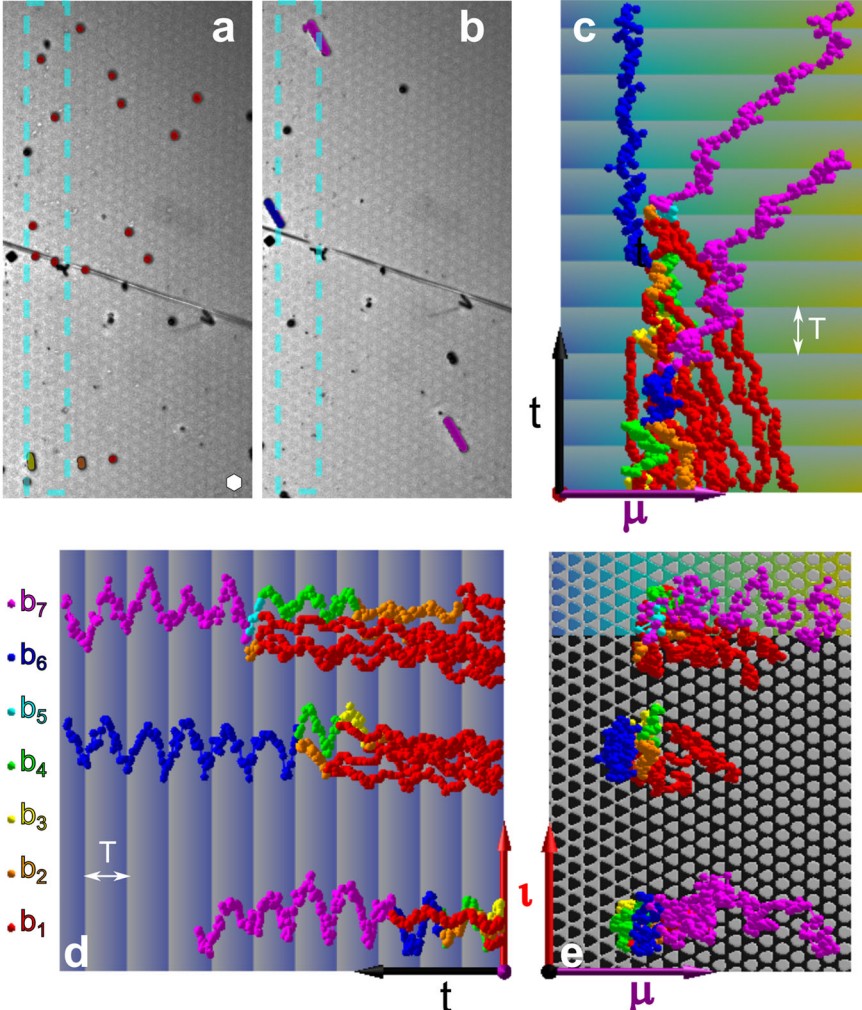

**Fig. 3 | $b_7$-synthesis.** Two reflection microscopy images at the beginning $t = 0$ (**a**) and after 7 loops $t \approx 7T$ (**b**) of the colloidal particles (colored according to the biped length) above the pattern subject to the loop $\mathcal{L} = \mathcal{L}_{entry} * \mathcal{L}_{exit,7}$. The hexagonal unit cell of the pattern (white hexagon) has lattice constant $a = 7$ μm. The active zone is marked in cyan. **c** Trajectories of single colloidal particles and bipeds (colored according to their length $b_n$) as a function of the time $t$ and the metamorphic coordinate. The background is colored according to the symmetry phase in the metamorphic direction, and the brightness of the background is periodic with the period $T$ of the modulation loop. **d** Trajectories of single colloidal particles and bipeds as a function of the time $t$ and the isomorphic coordinate. **e** Trajectories of single colloidal particles and bipeds as a function of the metamorphic coordinate and the isomorphic coordinate. The background shows the magnetization pattern of the magnetic thin film. Supplementary Video 7 of the synthesis of one $b_6$-biped and two $b_7$-bipeds is provided with the supplementary information.

particle control spaces are colored according to the location in action space and they are placed right above the corresponding region in action space. A larger gray copy of the control space simultaneously explaining the topologically protected transport anywhere in the pattern is attached as well.

For each symmetry phase, the white and black segmented fence in the control space $\mathcal{C}$ separates colored regions on the concave side of the fence from gray excess regions on the convex side of the fence. External magnetic fields pointing into the colored regions of $\mathcal{C}$ cause a colloidal potential with one minimum per unit cell in $\mathcal{A}$, while fields pointing into the gray excess regions of $\mathcal{C}$ cause a colloidal potential with two minima per unit cell in $\mathcal{A}$. The cusps of the fences where black and white segments meet are $\mathcal{B}_-$-bifurcation points. The cusps of the fences where the same colored segments meet are $\mathcal{B}_0$-bifurcation points. The fence line in $\mathcal{C}$ depends on the symmetry phase $\varphi$. For a $C_6$-like pattern ($\varphi = 60\pi/180$, magenta) there is one polar gray excess region fenced by twelve (white and black) segments of the fence that connect twelve bifurcation points. When decreasing the symmetry phase to $\varphi = 40\pi/180$ (bright blue), three further diamond-shaped excess regions disjoin from this polar excess region and move toward the equator of control space when the symmetry phase reaches the cyan $S_6$-symmetry. During the same decrease of the symmetry phase the polar excess region shrinks to a point and reappears at the opposite pole beyond the $S_6$-symmetric phase, recollects the diamond-shaped excess regions ($\varphi = 20\pi/180$, bright green) now in the opposite hemisphere to form a polar twelve segmented three-fold symmetric fence ($\varphi = 10\pi/180$, bright olive) that adopts the full six-fold symmetric polar shape ($\varphi = 0$, yellow) for the $\bar{C}_6$-like region in action space.

Consider an external field pointing toward a point in the colored one minimum region of control space and a colloidal particle occupying this one minimum in a unit cell of action space. When we redirect the external field in control space such that it crosses into the gray excess region, a new non-occupied (less deep) excess minimum and an excess saddle point form in the unit cell of action space. Let us call the two minima a white and a black minimum. If we enter the excess region of control space through a black (white) fence segment, the colloidal particle will occupy the black (white) minimum in action space while the excess minimum of the opposite color remains unoccupied. If one exits the gray excess region to the colored single minimum region of

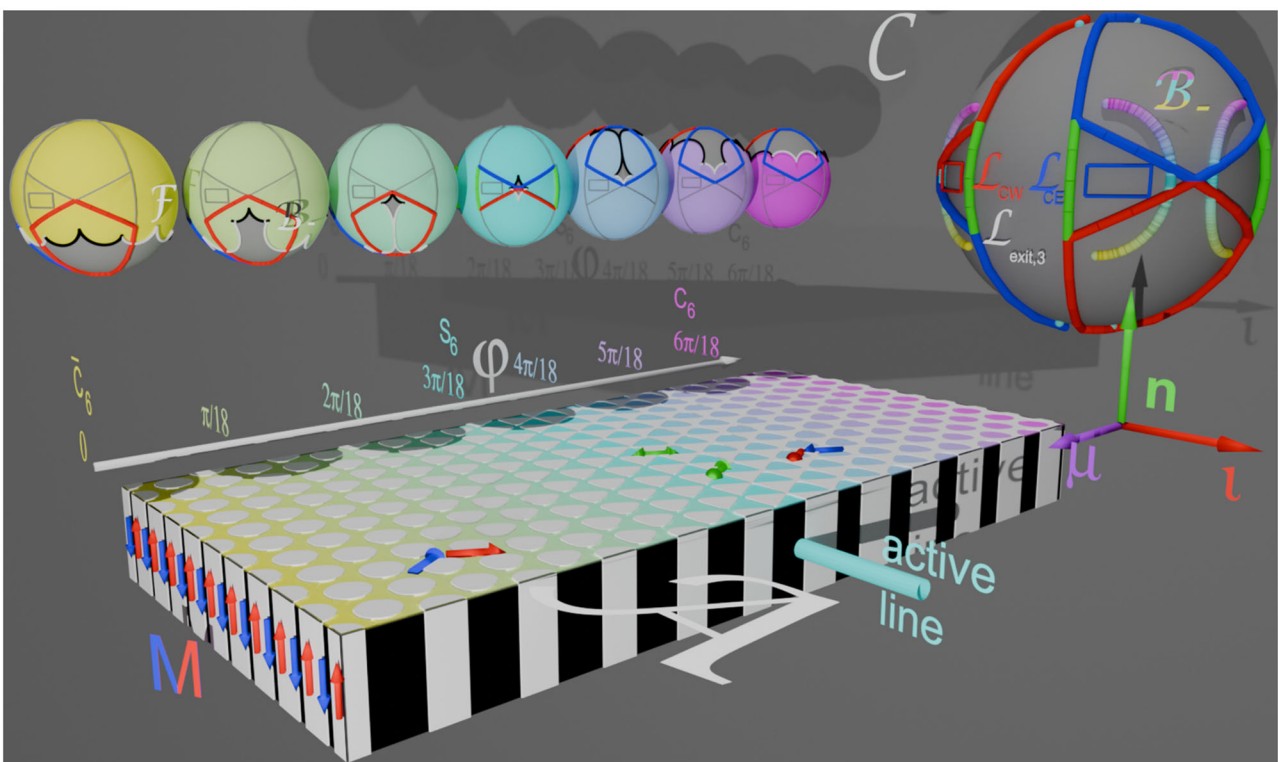

**Fig. 4 | Topology of the transport for single colloidal particles.** The magnetic pattern (action space $\mathcal{A}$) is colored according to the symmetry phase $\varphi$. On top of seven equally spaced regions, we place seven control spaces corresponding to the local symmetry phases in these regions. A larger gray copy of the control space summarizes what we learn from the single symmetry control spaces. It shows the (thicker) entry loop $\mathcal{L}_{entry}$ and the (thinner) exit loop $\mathcal{L}_{exit,3}$ used for the synthesis of $b_3$-bipeds. Both loops consist of sub-loops revolving in the mathematically positive (blue in the large copy of control space) and negative (red in the large copy of control space) sense. In the colored control spaces we show the positions of the corresponding fences $\mathcal{F}$ (white and black segments) for single colloidal particles of the corresponding symmetry phase. The fences are the external field orientations for which there are marginally stable potential minima in action space. The fences circulate around the gray excess regions where there are two minima per unit cell of the potential in action space. Cusps of the fences are bifurcation points. We distinguish $\mathcal{B}_-$-bifurcation points where black and white segments of the fence meet from $\mathcal{B}_0$-bifurcation points where similarly colored fence segments meet. Loops are colored gray in the small control spaces if they do not wind around $\mathcal{B}_-$-bifurcation points of the particular symmetry phase. These loops are topologically trivial in this region of action space. The transport direction of active topologically non-trivial loops is depicted by arrows in action space $\mathcal{A}$ using the same color as the loop. In the larger gray copy of the control space $\mathcal{B}_-$-bifurcation points join to (thick magenta-cyan-yellow) bifurcation lines when plotted as a function of the symmetry phase $\varphi$. The winding of a loop around a set of $\mathcal{B}_-$-bifurcation points determines the induced transport direction for the particular symmetry phase. As one moves through action space in the metamorphic direction $\boldsymbol{\mu}$ one eventually passes through the active (iso-phase) line. At the same time bifurcation points in control space pass from the northern part of the loop revolving in the mathematical positive (negative) sense to the southern part of the loop revolving in the opposite sense. The loops are eigen loops of the $\sigma_t$ mirror operation and, therefore, suppress any net motion into the isomorphic $\boldsymbol{\iota}$-direction. The entry loop, therefore transports single colloids and small bipeds toward the active zone on the $S_6$ isomorphic line of the pattern. The exit loop $\mathcal{L}_{exit,3}$ does not revolve around any of the $\mathcal{B}_-$-bifurcation points and, therefore does not transport single colloids. Supplementary Video 8 shows this figure from various perspectives.

control space via a similarly colored fence segment, the process is adiabatic, and the colloidal particle in action space remains in the same (or an equivalent) minimum as it resided upon entry of the excess region. If we exit the excess region of control space via a fence segment of opposite color, the occupied minimum disappears upon exit, and our colloidal particle performs an irreversible ratchet jump in action space toward the remaining minimum of opposite color in the same or a neighboring unit cell. There are, therefore adiabatic modulation loops that enter and exit the excess region of control space via similarly colored fence segments and irreversible ratchet loops that enter and exit via oppositely colored fence segments[10]. Black and white fence segments join in the control space at the $\mathcal{B}_-$-bifurcation points. An adiabatic (ratchet) loop must encircle an even (odd) number of $\mathcal{B}_-$-bifurcation points. Adiabatic loops that encircle a non-zero number of $\mathcal{B}_-$-bifurcation points are topologically non-trivial. Such non-trivial loops connect the one minimum in one unit cell to the one minimum of a neighboring unit cell. In order to predict the motion of a single colloidal particle it suffices to know the winding numbers of modulation loops in control space around the $\mathcal{B}_-$-bifurcation points. We, therefore, omit the fences in the larger gray copy of the control space

and just plot the family of the $\mathcal{B}_-(\varphi)$-bifurcation points, colored according to the symmetry phase of the pattern.

We use one (blue) loop on the northern hemisphere $\mathcal{L}_{NE}$ circling around two $\mathcal{B}_-(\varphi)$-bifurcation points for a subset of symmetry phases $\varphi > 30\pi/180$ (blue-magenta). The loop $\mathcal{L}_{NE}$ is topologically non-trivial for this subset. After completion, the loop transports single colloids from the $C_6$-like region toward the $S_6$-symmetric region, provided that we choose the proper sense of circulation. The proper sense of circulation is found by using the rolling wheel rule: Consider a virtual wheel rolling on the pattern. The axis of the wheel is given by the averaged direction of the two $\mathcal{B}_-$ bifurcation points which the loop winds around. The direction of motion of a single colloidal particle is perpendicular to this axis and is provided by the direction that the virtual wheel rolling on the pattern below it would take if spinning around the axis with the same winding number as the loop. In action space $\mathcal{A}$ of Fig. 4 we depict the direction of transport of this loop for $\varphi > 30\pi/180$ by arrows with the same color as the loop. The loop $\mathcal{L}_{NE}$ is topologically trivial for colloids lying on the other side of the $S_6$-symmetry line in action space $\mathcal{A}$, ($\varphi < 30\pi/180$, yellow-green). The loop, therefore, causes no transport of colloids on this part of the action

space. We therefore color $\mathcal{L}_{NE}$ in the single symmetry phase control spaces of this region in gray.

In order to transport single colloidal particles from the other side of the active $S_6$-symmetry line, we use a second (red) south-eastern, loop $\mathcal{L}_{SE} = \sigma_n(\mathcal{L}_{NE})$ being a mirror image of the north-eastern loop mirrored at the equatorial plane. $\mathcal{L}_{SE}$ lies on the southern hemisphere and circulates in the opposite sense around the $\mathcal{B}_-(\varphi)$-bifurcation points beyond the $S_6$-symmetry ($\varphi < 30\pi/180$, yellow-green). In the region $\varphi > 30\pi/180$ (blue-magenta) $\mathcal{L}_{SE}$ is topologically trivial (and therefore colored gray in the corresponding control spaces). The southeastern loop therefore, transports single colloids in $\mathcal{A}$ on the other side of the $S_6$ symmetric active line in the opposite direction. As a result, the double loop transports all single colloids toward the $S_6$ symmetric line and therefore, increases the density of colloids at this line. Two further western loops $\mathcal{L}_{NW} = \sigma_t(\mathcal{L}_{NE})$ and $\mathcal{L}_{SW} = \sigma_t(\mathcal{L}_{SE})$ are mirrored at the isomorphic $\iota$-plane (see Fig. 1). Together with the eastern loops they suppress any net motion into the isomorphic $\iota$-direction. Similarly to the other loops in the fixed symmetry phase control spaces, we color the eastern loop with their sense of winding in the regions where they are topologically non-trivial and gray otherwise. We call the concatenation of the four loops the entry loop $\mathcal{L}_{entry} = \mathcal{L}_{NE}{}^*\mathcal{L}_{SE}{}^*\mathcal{L}_{NW}{}^*\mathcal{L}_{SW}$.

The green loop segments of the concatenated eastern loop $\mathcal{L}_{NE}{}^*\mathcal{L}_{SE}$ expand the loop with a ratchet triangle. Instead of using the original loops, we deform one of the diagonals of $\mathcal{L}_{NE}{}^*\mathcal{L}_{SE}$ to take the detour via the green segments and the other diagonal. This extended loop is topologically equivalent to the undeformed loop for symmetry phases far away from the $S_6$ symmetry but alters the winding numbers around the $\mathcal{B}_-$ bifurcation points for symmetry phases close to the $S_6$ symmetry. In the experiments we alternately deform one and the other diagonal of the loops in this way such that winding numbers around the $\mathcal{B}_-$ bifurcation points in the (cyan) active zone at the $S_6$ symmetry line are odd. This ensures that single particles do not come to rest in the $S_6$ region of the active line in action space $\mathcal{A}$ but hop back and forth in a ratchet motion. We depict this back-and-forth hopping in action space with green double-sided arrows in Fig. 4. We also show figure 4 from various perspectives in our Supplementary Video 8.

For sufficient colloidal density, the probability of dipolar interactions between the colloids polymerizing several colloids to a biped $b_n$ consisting of $n$ single colloids becomes significant. The $S_6$-symmetry line serves as a polymerization initiation site. Bipeds grow via single colloid addition polymerization but also via the addition of two bipeds. We use a central equatorial exit loop $\mathcal{L}_{exit} = \mathcal{L}_{CE}{}^*\mathcal{L}_{CW}$ consisting of two mirror symmetric loops $\mathcal{L}_{CE}$, and $\mathcal{L}_{CW} = \sigma_t(\mathcal{L}_{CE})$ circling around two mirror symmetric points on the equator of control space $\mathcal{C}$ commanding outgrown bipeds of the desired length to walk away from the $S_6$-symmetry line. The loop is chosen to be trivial for juvenile bipeds and single colloids at any location on the pattern. The walking direction for the outgrown bipeds for each of the loops $\mathcal{L}_{CE}$ and $\mathcal{L}_{CW}$ is perpendicular to the external field vector pointing to the encircled point of each of the loops and the length selection can be made via the proper placement of those points. In Fig. 4, we show the exit loop $\mathcal{L}_{exit,3}$ in control space and it can be seen that it does not wind around any of the $\mathcal{B}_-(\varphi)$-bifurcation points of single colloidal particles. The exit loop is thus trivial with respect to single colloids and does not affect the net motion of single colloidal particles. For understanding the behavior of bipeds $b_n$ consisting of more than one colloidal particle, it is useful to introduce the polydirectional transcription space $\mathcal{T}_p$ that is the vector space of the biped end to end vectors[8].

## Juvenile and outgrown bipeds response

The orientation of the (dipolar) biped is locked to that of the external field with the northern foot being a magnetic north pole and the

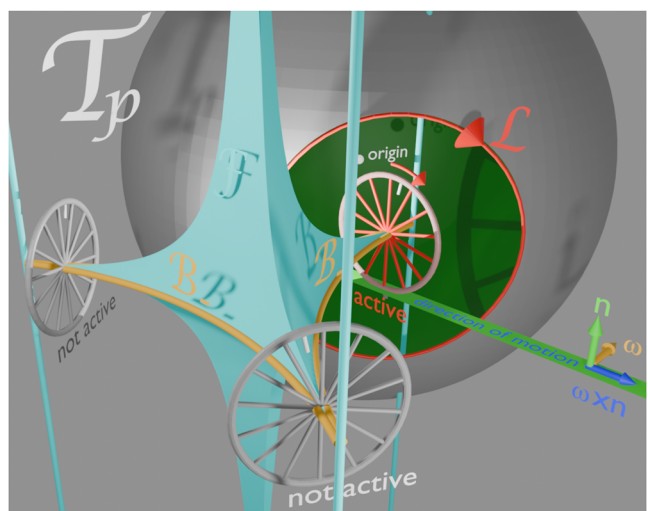

**Fig. 5 | Determining the direction of motion by using the rotating wheel rule.** Shown in the figure is one cyan fence pocket in transcription space separating biped vectors outside the fence supporting one minimum position of a biped per unit cell from vectors inside the fence supporting two minima. Three pairs of orange $\mathcal{B}_-$ bifurcation lines define the axes of three *virtual* wheels rolling on *virtual* support. One of the wheels, -- the red wheel -- is activated by a red loop circulating one pair of bifurcation lines in the mathematical negative sense. The biped will be transported in the same direction as the activated wheel would roll. The activated wheel shares its winding number ($w = -1$) with the activating loop $\mathcal{L}$. Any loop $\mathcal{L}$ having the same winding number will result in the same transport. The path of the loop is on a sphere of radius of the length $b_n$ of the corresponding biped. The exact path of the loop does not matter. The loop's topological invariant, the winding number, does matter.

southern foot being a south pole. Let $\mathbf{b}_n$ denote the vector from the northern foot to the southern foot of a biped of length $b_n$. Because of the locking of the direction of $\mathbf{b}_n$ to the external field $\mathbf{H}_{ext}$ we can, instead of depicting the fence in control space $\mathcal{C}$, print the same fence on a sphere of radius $b_n$. If we continuously vary the biped length, the fence lines of different length bipeds join on neighboring concentric spheres to form a fence surface $\mathcal{F}$ in $\mathcal{T}_p$.

In Fig. 5, we outline the rotating wheel rule for bipeds that when applied to the other figures of this section, will tell us the direction of motion of the bipeds of different lengths in one particular region of action space $\mathcal{A}$. The figure shows the transcription space of possible end to end vectors $\mathbf{b}$, in which we compute the cyan locations of marginable stable end to end vectors, called the fence $\mathcal{F}$, that consists of concave areas separated by orange bifurcation lines $\mathcal{B}_-$. Two bifurcation lines $\mathcal{B}_-$ define an axis $\boldsymbol{\omega}$ on which we can imagine a virtual wheel. In Fig. 5 three virtual wheels sit on three pairs of $\mathcal{B}_-$ bifurcation lines. A loop circulating around one of the three axes will activate the wheel sharing the same winding number with the activating loop. The direction the wheel would roll on a virtual support below the wheel gives us the direction $\boldsymbol{\omega} \times \boldsymbol{n}$ of motion of the biped. Note that if the biped is not a single colloidal particle, the loop must not wind around the fence in a symmetric way. Sharing the winding number, – a topological invariant – with the wheel is sufficient to drive the virtual wheel forward. The rotating wheel rule is sufficient to allow the reader to predict the direction of motion. The location of the fences and bifurcation lines are computed numerically.

In previous work[8] we have shown that for fixed symmetry phase $\varphi$ this fence surface is periodic and repeats in transcription space $\mathcal{T}_p$ with the primitive unit vectors of the magnetization pattern in action space $\mathcal{A}$. In Fig. 6 we plot the fence surface inside the Wigner Seitz cell with the origin $\mathbf{b}_n = \mathbf{0}$ in the center. The fence consists of surface

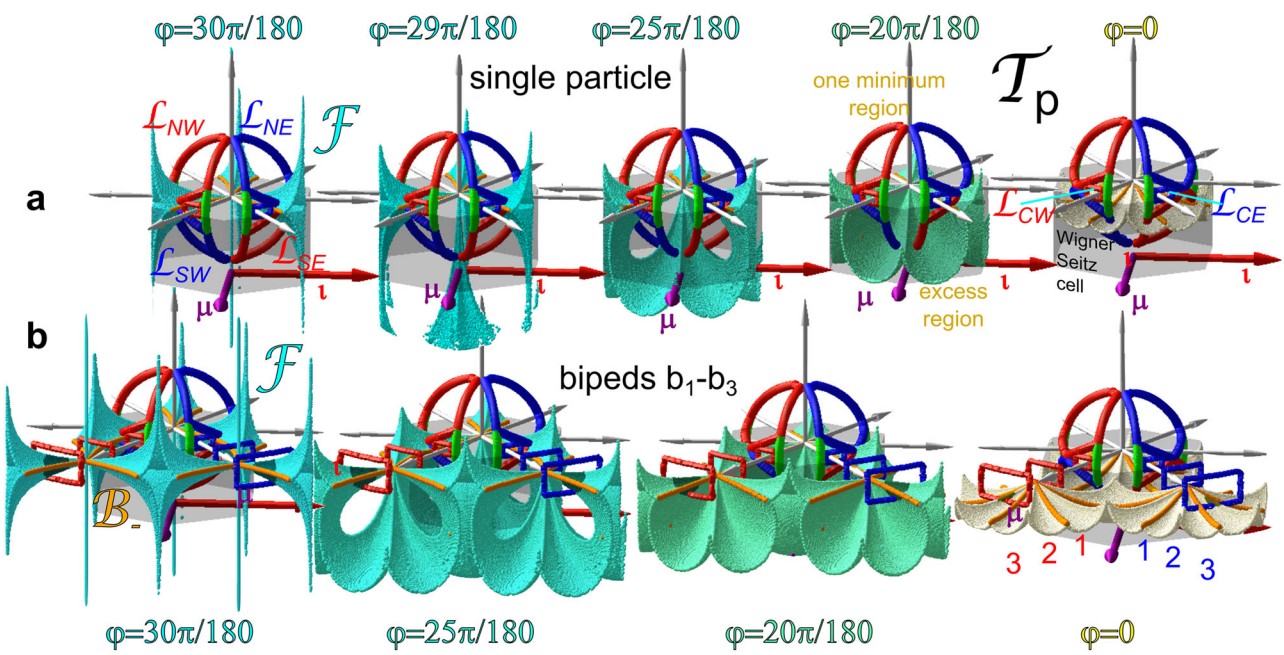

**Fig. 6 | Topological character of the loops in transcription space. a** Effect of the entry and exit loops $\mathcal{L}_{exit,3}$ for small bipeds including single colloidal particles in the central Wigner Seitz cell of transcription space for different symmetry phases. The southern sub-loops $\mathcal{L}_{SW}$ and $\mathcal{L}_{SE}$ are topologically non-trivial for small symmetry phases and wind around two of $\mathcal{B}_-$ bifurcation lines (orange) of the fences. For larger symmetry phase the loops poke through the fence lobes of trigonal bipyramid fences and wind around none of $\mathcal{B}_-$ bifurcation lines. If one extends the southeastern loop with one ratchet triangle, the extended loop winds around one $\mathcal{B}_-$ bifurcation line and causes non-trivial ratchet jumps of the single colloids in the active zone back and forth into the metamorphic $\pm\,\boldsymbol{\mu}$-direction. The fences for $\varphi > 30\pi/180$ (not shown) are mirror reflections at the $b_z = 0$-plane and cause the northern sub entry loops having opposite sense of revolution to transport single colloids into the opposite direction as the fences for $\varphi < 30\pi/180$. Both exit sub-loops $\mathcal{L}_{CE}$ and $\mathcal{L}_{CW}$ are topologically trivial for the single colloids and do not wind around any of the fences. **b** Periodic continuation of the fences in an extended zone scheme together with exit loops for bipeds of length $b_1, b_2,$ and $b_3$. Exit loops for $n < 3$ are trivial, while the exit loops for $b_3$ are topologically non-trivial and wind around (shaded) lobes of the fences for large symmetry phases. For smaller symmetry phases, the exit loops still wind around the same $\mathcal{B}_-$ bifurcation lines (orange), and the loop pokes through equivalent fence facets with the minimum upon entry of the excess volume remaining stable when the loop re-exits into the one minimum volume. The non-trivial exit loop for $b_3$ causes the $b_3$-bipeds to walk into the metamorphic direction irrespective of their position on the metamorphic pattern in action space $\mathcal{A}$.

segments of positive Gaussian curvature that join at bifurcation lines of infinite negative Gaussian curvature. We show the fences for the symmetry phase $\varphi = 30\pi/180, 29\pi/180, 25\pi/180, 20\pi/180,$ and $0\pi/180$. The fences for $\sigma_n(\varphi) = 60\pi/180 - \varphi$ (not shown) are obtained by taking the mirror image $\mathcal{F}_{60\pi/180-\varphi} = \sigma_n(\mathcal{F}_\varphi)$ of the fence $\mathcal{F}_\varphi$ at the $b_z = 0$ plane. The fence in transcription space $\mathcal{T}_p$ separates regions with one minimum per unit cell in $\mathcal{A}$ of the biped colloidal particle potential on the concave side of the fence in $\mathcal{T}_p$ from excess regions with two minima per unit cell in $\mathcal{A}$ of the pattern on the convex side of the fence in $\mathcal{T}_p$. The fence in $\mathcal{T}_p$ for the $S_6$ symmetric situation ($\varphi = 30\pi/180$) consists of a trigonal bipyramid pocket located at one of the edges of the Wigner Seitz cell harboring two minima of the potential and a line through the origin in $b_z$-direction having zero volume just failing to contain two minima (or maxima) since its volume is zero. Three faces of the bipyramid join at the apex located at $b_z \to \infty$ and three faces join at the antapex located at $b_z \to -\infty$ of the bipyramid. When we slightly reduce the symmetry phase to e.g., $\varphi = 29\pi/180$ the apex of the bipyramid retreats to a finite value of $b_z$. Both the zero volume line and the trigonal bipyramid open at their antapexes and connect to form a surface separating a region below with two minima from the region above with one minimum. The region of one minimum for $30\pi/180 > \varphi > 20\pi/180$, therefore, is not simply connected but there are connections through the holes between the base of the trigonal bipyramid and the lower part of the surfaces. A second topological transition at $\varphi = 20\pi/180$ closes these holes to separate the now simply connected region of one minimum from a simply connected region with two minima. For $\varphi = 0$, the three-fold symmetry of the fence around the origin augments to a six-fold symmetry.

A biped control loop $\mathcal{L}_n = \frac{b_n}{H_{ext}}\mathcal{L} \subset \mathcal{T}_p$ is the transcription of the loop $\mathcal{L} \subset \mathcal{C}$ from control space $\mathcal{C}$ to a spherical surface in transcription space of radius $b_n = nd$. In the upper part of Fig. 6, we show the transcripted entry and exit loop $\mathcal{L}_{entry}$ and $\mathcal{L}_{exit,3}$ for a single colloid $n = 1$ together with the fence surface. The cut of the sphere of radius $b_1$ with the fences repeats what has been shown in Fig. 4. The entry loop circulates the fences for the low symmetry phase but pokes through the fences when the symmetry phase comes close to $\varphi = 30\pi/180$. The exit loop does not wind around any fence surface and is topologically trivial for $n = 1$. When we periodically continue the fences into neighboring Wigner Seitz cells, we can infer the effect of the exit loops of bipeds $b_n = nd$ with $n = 1, 2,$ and $n = 3$.

The transcriptions of the exit loop $\mathcal{L}_{exit,3}$ for $n = 1$ and $n = 2$ do not wind around any fence surfaces, while for $n = 3$ it winds around two of the (shaded) base handles of the trigonal bipyramids located in unit cells away from the origin and therefore non-trivially transports $b_3$-bipeds perpendicular to the two fence handles or the surfaces developing from the handle for symmetry phases $\varphi < 30\pi/180$. The symmetry of the two sub-loops of the exit loop is such as to cancel the transport into the isomorphic $\boldsymbol{\iota}$-direction, and only the transport along the metamorphic $\boldsymbol{\mu}$-direction remains. The exit loop remains invariant under reflection at the $b_z = 0$-plane followed by a time reversal operation and therefore, has the same effect in regions of action space $\mathcal{A}$ where $60\pi/180 > \varphi > 30\pi/180$. Bipeds of length $b_3$ are therefore, globally transported the same way on the entire metamorphic pattern.

### Changing the outgrown length

In Fig. 7 we show how repositioning the exit loop $\mathcal{L}_{exit,3} \to \mathcal{L}_{exit,7}$ changes the length of the outgrown biped from $b_3$ to $b_7$. The

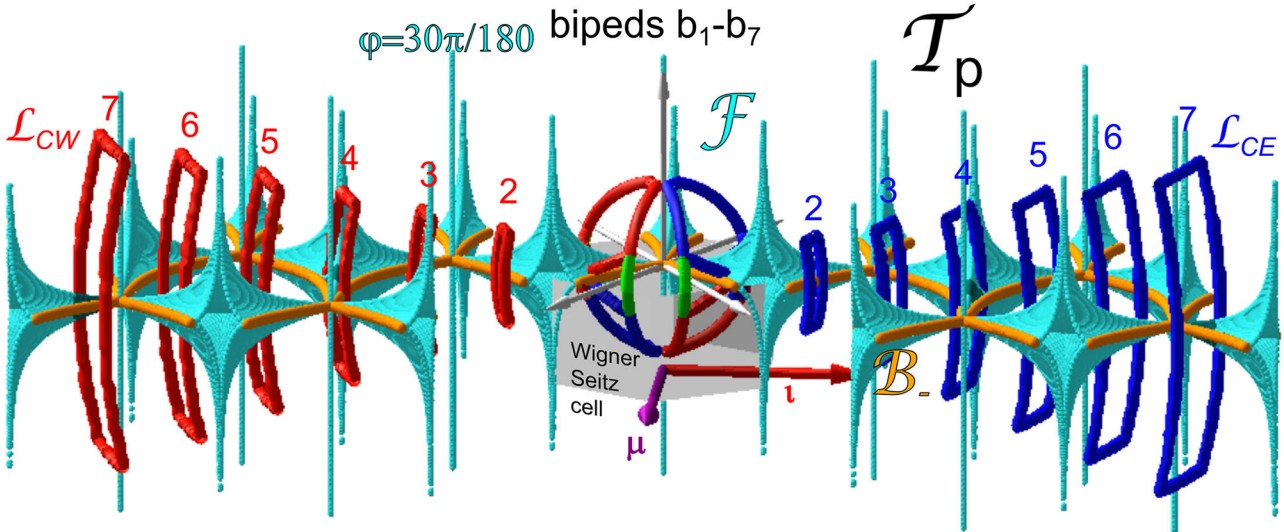

**Fig. 7 | Effect of repositioning the exit loop.** For an exit loop $\mathcal{L}_{exit,7}$ repositioned with respect to the exit loop $\mathcal{L}_{exit,3}$ of Fig. 6 and a symmetry phase of $\varphi = 30\pi/180$, the loop is trivial for $b_1$, $b_2$, $b_3$, $b_5$, and $b_6$. The motion of both sub-loops is non-trivial for $b_4$, but the motion is canceled by the $\sigma_\iota$-symmetry of both sub-loops. The first transcribed exit loops $\frac{b_n}{H_{ext}^{ext}}\mathcal{L}_{exit,7}$ length to be transported into the metamorphic $\mu$-direction is a biped $b_7$ with the exit loop winding around the shaded lobes of the outermost handles of the periodically extended fence trigonal bipyramids. The effect is topologically robust when the $b_7$-biped walks into regions of lower symmetry phase $\varphi$.

transcribed exit loops $\frac{b_n}{H_{ext}^{ext}}\mathcal{L}_{exit,7}$ in transcription space $\mathcal{T}_p$ for synthesizing a $b_7$ biped in Fig. 7 do not wind around any fences for $n = 1, 2, 3, 5$, and 6. For $\varphi = 30\pi/180$ the $b_4$ transcribed exit loop winds around two handles of two different trigonal bipyramids, both oriented perpendicular to the isomorphic $\iota$-direction. Each winding of the two sub-loops individually transports $b_4$-bipeds into the isomorphic $\iota$-direction but the two sub-loops cancel out any motion with the winding around the second handle undoing the motion caused by the winding around the first (see also the green experimental $b_4$-trajectories in Fig. 3d). The first bipeds to experience a non-trivial topological transport are $b_7$-bipeds. The transcribed exit loop for $b_7$ winds around the two outermost handles of two different trigonal bipyramids, depicted and shaded in Fig. 7. The orientations of both handles are such that the normal in-plane vectors to both handles have a metamorphic component. The two sub-loops – like for the $b_4$-case – cancel any motion into the isomorphic $\iota$-direction but they add up to cause motion into the metamorphic $\mu$-direction. The topology of the winding remains robust as the symmetry phase $\varphi$ changes to lower values as the $b_7$-biped walks away from the active line.

## Discussion

The synthesis or assembly of a product can occur via thermodynamic driving forces, in which case we talk about self assembly[16–18]. The directed assembly[19–21] via an externally controlled addition of components via joining micro fluidic channels on a lab on a chip is an externally enforced alternative to the creation of products that are not necessarily in thermodynamic equilibrium. Our topological synthesis is an example of the latter strategy but using motion[22–24] of the educts that actively self assemble to the final product. In contrast to conventional lab on the chip devices[25], our approach offers the flexibility of using the same pattern imprinted on the device for the production of alternative products. The flexibility arises via the ability to determine the outgrown product length via a smart choice of the applied external modulation. The choice to be an outgrown product, however, is an active and robust choice of the colloidal biped itself allowing the synthesis to be made without external control of its success. The topological nature of the internal decision made by the products of our device is an ingredient shared with many bio-synthetic processes in vivo.

In fact, our device functions like an enzyme in biology. Due to the low concentration of colloids, no polymer addition reactions are supposed to occur outside the active zone. Polymer addition reactions are wanted at the active line, not elsewhere. By topologically transporting single colloids into the active zone, the colloidal concentration is increased there as to increase the reaction rate. The choice of concentration, therefore, is critical for the proper functioning of the device. Since single colloids are kept at the same distance of at least one unit cell, while transporting them to the active zone, polymer addition reactions of single colloids never happened outside the active zone in our experiments. However, once an outgrown biped leaves the active zone, it counter propagates to the single colloids, and unwanted addition reactions can elongate the length of bipeds beyond the outgrown length. It is possible to design more complex metamorphic patterns where outgrown bipeds may avoid further collisions upon exit and use more sophisticated driving loops. The current work just shows the potential of the method. It produces outgrown bipeds without polydispersivity. The kinetics is of a Michaelis-Menten type saturating at a concentration when all active zones are busy. This is the case once one section of the length $b_{n^*}$ of the active line yields roughly one outgrown biped per $n^*$ periods of the loop. Due to the requirement of adiabatic transport, our yield is low, however, our precision is high.

Theoretically, there is no size limit to the production of bipeds of any length. In practice, in the experiments, there is a size limit. The robustness of the process relies on loops encircling the fences by keeping a distance to the bifurcation locations. The number of bifurcation locations in transcription space increases with the number of bipeds (all transcription loops to bipeds with $n < n^*$ have to fulfill the distance requirement, compare Fig. 6 and 7) which becomes increasingly challenging. The adiabatic period $T \propto n^{*3/2}$ becomes longer and longer, and at some point, we would need to reduce the modulus of the morphing reciprocal vectors $\mu$ to avoid a biped having one foot in the $S_6$ the other in the $C_6$ region. Moreover, the total cross section of a biped grows proportional to its size, such the concentration of single colloids must be reduced to avoid unwanted collisions of the biped after its exit of the active zone.

Topological length control is a useful strategy for active self-assembly out of equilibrium. Adiabatic external modulation loops can

be adapted in a versatile way to let colloidal particles decide their final length by themselves. Here, we have shown the internal active self-assembly of magnetic colloids to a biped for six different outgrown lengths. Of course, the externally applied control loop guides the active motion toward the desired length. Once the external loop is applied, however, no further external quality control, whether the components assemble to the desired structures, is needed.

## Methods

### Pattern

The pattern is a thin Co/Au layered system with perpendicular magnetic anisotropy lithographically patterned via ion bombardment[26,27]. The metamorphic pattern lattice constant is $a = 7\,\mu m$, and the modulation period is $2\pi/\mu = 280\,\mu m$, see the sketch in Fig. 1a. The magnetic pattern is spin-coated with a $1.6\,\mu m$ polymer film that serves as a spacer between the magnetic film and the colloidal particles.

### External field

The uniform external magnetic field has a magnitude of $H_{ext} = 4\,kAm^{-1}$ (significantly smaller than the coercive field of the magnetic pattern), and it is generated by one vertical and four horizontal computer-controlled coils arranged around the sample at ninety degrees.

### Modulation loops

The loops consist of connections between points $P_{ij} = (\theta_i, \phi_j)$ given as spherical coordinates in $\mathcal{C}$, with $N = P_{1,j}$ and $S = P_{-1,j}$ the north and south pole. For the entry loop, we use the angles of Table 1, as well as the mirror tilt angles $\theta_{-i} = \sigma_n(\theta_i) = \pi - \theta_i$, mirrored at the equatorial plane, and the azimuthal angles $\phi_{-i} = \sigma_t(\phi_i) = -\phi_i$, mirrored at the $t$-plane. The exit loop angles are tabulated in Table 2 according to the value $n^*$ of the outgrown biped to be synthesized.

The sequence of points of the concatenations of all but the loops with $n^* = 2$ or $n^* = 4$ loops are:

$$P_{0,2}, P_{-3,3}, P_{2,3}, P_{2,2}, N, P_{4,5}^{n^*}, P_{4,4}^{n^*}, P_{-4,4}^{n^*}, P_{-4,6}^{n^*}, P_{4,6}^{n^*}, P_{4,5}^{n^*},$$
$$N, P_{2,-2}, P_{2,-1}, P_{3,-1}, P_{0,-2}, P_{-3,-3}, P_{-2,-3}, P_{-2,-1},$$
$$P_{3,-1}, P_{0,-2}, P_{-3,-3}, P_{2,-3}, P_{2,-2}, N, P_{4,-5}^{n^*}, P_{4,-4}^{n^*}, P_{-4,-4}^{n^*},$$
$$P_{-4,-6}^{n^*}, P_{4,-6}^{n^*}, P_{4,-5}^{n^*}, N, P_{2,2}, P_{2,1}, P_{3,1}, P_{0,2}, P_{-3,3}, P_{-2,3},$$
$$P_{-2,1}, P_{3,1}, P_{0,2}, P_{3,3}, P_{-2,3}, P_{-2,2}, S, P_{-4,5}^{n^*}, P_{-4,4}^{n^*}, P_{4,4}^{n^*},$$
$$P_{4,6}^{n^*}, P_{-4,6}^{n^*}, P_{-4,5}^{n^*}, S, P_{-2,-2}, P_{-2,-1}, P_{-3,-1}, P_{-0,-2},$$
$$P_{3,-3}, P_{2,-3}, P_{2,-1}, P_{-3,-1}, P_{0,-2}, P_{3,-3}, P_{-2,-3}, P_{-2,-2},$$
$$S, P_{-4,-5}^{n^*}, P_{-4,-4}^{n^*}, P_{4,-4}^{n^*}, P_{4,-6}^{n^*}, P_{-4,-6}^{n^*}, P_{-4,-5}^{n^*}, S, P_{-2,2},$$
$$P_{-2,1}, P_{-3,1}, P_{0,2}, P_{3,3}, P_{2,3}, P_{2,1}, P_{-3,1}, P_{0,2}$$

### Table 1 | Entry loop angles

| $\theta_0$ | $\theta_1$ | $\theta_2$ | $\theta_3$ | $\phi_1$ | $\phi_2$ | $\phi_3$ |
|---|---|---|---|---|---|---|
| $\pi/2$ | 0 | $\pi/18$ | $7\pi/18$ | $\pi/9$ | $\pi/3$ | $5\pi/9$ |

### Table 2 | Exit loop angles

| $n^*$ | $\theta_4^{n^*}$ | $\phi_4^{n^*}$ | $\phi_5^{n^*}$ | $\phi_6^{n^*}$ |
|---|---|---|---|---|
| 2 | $70\pi/180$ | $140\pi/180$ | $170\pi/180$ | $170\pi/180$ |
| 3 | $85\pi/180$ | $25\pi/180$ | $35\pi/180$ | $45\pi/180$ |
| 3&4 | $70\pi/180$ | $100\pi/180$ | $100\pi/180$ | $140\pi/180$ |
| 5 | $75\pi/180$ | $5\pi/180$ | $20\pi/180$ | $35\pi/180$ |
| 6 | $75\pi/180$ | $125\pi/180$ | $135\pi/180$ | $135\pi/180$ |
| 7 | $75\pi/180$ | $65\pi/180$ | $75\pi/180$ | $75\pi/180$ |

For $n^* = 2$ the effect of $\mathcal{L}_{entry}$ on $b_2$-bipeds is topologically equivalent to that on single colloidal particles. Therefore the entry loop transports $b_2$-bipeds toward the active zone. To overcome this transport, we duplicate the winding numbers of the exit loops according to $\mathcal{L}_{exit,2} \rightarrow \mathcal{L}_{exit,2}^2$, that makes the exit loop dominant over the entry loop for the $b_2$-bipeds. For the point sequence one should replace all exit segments according to the scheme $N, P_{4,5}^2, P_{4,4}^2, P_{-4,4}^2, P_{-4,6}^2, P_{4,6}^2, P_{4,5}^2, N \rightarrow N, P_{4,5}^2, P_{4,4}^2, P_{-4,4}^2, P_{-4,6}^2, P_{4,6}^2, P_{4,5}^2, P_{4,4}^2, P_{-4,4}^2, P_{-4,6}^2, P_{4,6}^2, P_{4,5}^2, N$ and so forth.

For $n^* = 4$ there are no simple robust exit loops not affecting bipeds $b_n$ with $n < 4$. We, therefore, apply two exit loops and replace a non-robust exit loop (that exists) by a robust one according to $\mathcal{L}_{exit,4} \rightarrow \mathcal{L}_{exit,3\&4} * \mathcal{L}_{exit,3}^{-1}$, where the first robust loop $\mathcal{L}_{exit,3\&4}$ transports both, $b_3$ and $b_4$, into the metamorphic direction followed by the second loop $\mathcal{L}_{exit,3}^{-1}$ transporting $b_3$ against the metamorphic direction. For the point sequence one should replace all exit segments according to the scheme $N, P_{4,5}^{n^*}, P_{4,4}^{n^*}, P_{-4,4}^{n^*}, P_{-4,6}^{n^*}, P_{4,6}^{n^*}, P_{4,5}^{n^*}, N \rightarrow N, P_{4,5}^{3\&4}, P_{4,4}^{3\&4}, P_{-4,4}^{3\&4}, P_{-4,6}^{3\&4}, P_{4,6}^{3\&4}, P_{4,5}^{3\&4}, N, P_{4,5}^3, P_{4,6}^3, P_{-4,6}^3, P_{-4,4}^3, P_{4,4}^3, P_{4,5}^3, N$ and so forth.

The point sequences for all outgrown bipeds include transfer segments between the eastern and western parts of the entry loop and between the exit and entry loop that we have not shown in the main part of the work. Such transfer segments are topologically trivial and irrelevant to understanding the topological invariants governing the transport. In principle the transfer between loops can be made anywhere, however, the equatorial region should not be used due to increased friction of a biped with the solid support. Our connectors between loops are all via the north $N = P_{1,j}$ or south pole $S = P_{-1,j}$.

The adiabatic nature is ensured by keeping the Mason number of all bipeds below one $\mathcal{M} \ll 1$. The Mason number of an outgrown biped is $\mathcal{M} = 2\pi\eta n^{*3/2}/\mu_0\chi_{eff}H_{ext}H_pT_F$ (with $\eta$ the shear viscosity of the fluid and $\chi_{eff}$ the effective magnetic susceptibility of a colloidal particle, and $T_F$ the period of a fundamental loop.). The effect of non-adiabatic driving has been discussed in ref. 15. In our experiments with $n^* = 7$ we used a modulation period of $T_F = 25\,s$ for a fundamental loop. Our modulation loop consists of twelve fundamental loops such that the largest period used was $T \approx 5$ min for our largest outgrown biped.

### Visualization

The colloids and the pattern are visualized using reflection microscopy. The pattern is visible because the ion bombardment changes the reflectivity of illuminated regions as compared to that of the masked regions. A camera records video clips of the single colloidal particles and the bipeds.

### Fence equations

A biped is subject to a total potential proportional to $-\int_{biped} \mathbf{d^3r}\, \mathbf{H}_{ext} \cdot \mathbf{H}_p$, which is the integral over the biped volume of the coupling between the external $\mathbf{H}_{ext}$ and the pattern $\mathbf{H}_p = -\nabla\psi$ fields[11,12]. This coupling leads to an effective biped potential $V$ proportional to the difference in magnetostatic potential at the two feet. That is,

$$V(\mathbf{r}_\mathcal{A} + z\mathbf{n}, \mathbf{b}, \varphi) \propto \psi(\mathbf{r}_\mathcal{A} + \mathbf{b}/2, \varphi) - \psi(\mathbf{r}_\mathcal{A} - \mathbf{b}/2, \varphi) \quad (3)$$

with the biped centered at $\mathbf{r}_\mathcal{A}$ and with $\psi(\mathbf{r}, \varphi) \propto e^{-qz}\sum_{i=0}^2 \cos(\mathbf{q_i} \cdot \mathbf{r} - \varphi)$ the magnetostatic potential. Note that $V$ depends explicitly on $\mathbf{H}_{ext}$ via the one-to-one correspondence between $\mathbf{b}/b$ and $\mathbf{H}_{ext}/H_{ext}$. Transport of a biped $b_n$ after completion of one modulation loop $\mathcal{L} \subset \mathcal{C}$ occurs provided that $\mathcal{L}_n = \frac{b_n}{H_{ext}}\mathcal{L} \subset \mathcal{T}_p$ winds around bifurcation lines, which are the cusps of the fences. The fences are those orientations $\mathbf{b} \in \mathcal{T}_p$ for which the potential is marginally stable[11], i.e., the set of biped orientations for which $\nabla_\mathcal{A}V = 0$ and $\det(\nabla_\mathcal{A}\nabla_\mathcal{A}V) = 0$. For the present metamorphic pattern and for a slowly varying symmetry phase $|\nabla_\mathcal{A}\varphi| \ll q$ both conditions have been numerically determined to be fulfilled along

the fence surfaces in $\mathbf{b} \in \mathcal{T}_p$ depicted in Fig 6. For a constant symmetry phase the biped potential is periodic and invariant under the simultaneous transformation $\mathbf{b} \rightarrow \mathbf{b} + \mathbf{a}_i$ and $\mathbf{r}_{\mathcal{A}} \rightarrow \mathbf{r}_{\mathcal{A}} + \mathbf{a}_i/2$ with $\mathbf{a}_i$, $i = \{0, 1, 2\}$, a lattice vector, cf. Eq. (3). The same periodicity in $\mathbf{b} \in \mathcal{T}_p$ applies, therefore, to the fences.

## Data availability

All the data supporting the findings are available from the corresponding author. All trajectories of particles are extracted from Supplementary Videos 2–7. We append the tracked particle files and a maple code in a supplementary dataset called figurerawdata.zip that converts this data into figures 2 and 3. Source data are provided with this paper.

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

## Acknowledgements

T.M.F. and D.d.l.H. acknowledge funding by the Deutsche Forschungsgemeinschaft (DFG, German Research Foundation) under project number 531559581 and by the Open Access Publishing Fund of the University of Bayreuth. P.K., M.U., and F.S. acknowledge financial support from the National Science Centre Poland through the OPUS funding (Grant No. 2019/33/B/ST5/02013).

## Author contributions

J.E., F.F., D.d.l.H., and T.M.F. designed and performed the experiment, computed the fences and bifurcation points and lines, and wrote the manuscript with input from all the other authors. J.E. and F.F. contributed equally to this work. P.K., M.U., and F.S. produced the magnetic film. S.A., & A.r.E. performed the fabrication of the micromagnetic metamorphic patterns within the magnetic thin film.

## Funding

## Competing interests

The authors declare no competing financial or non-financial interests.
