## [Peer Review File · Nature Communications]

REVIEWER COMMENTS

Reviewer #2 (Remarks to the Author):

This paper investigates the migration and assembly of paramagnetic colloids moving above a planar substrate due both to a homogeneous, time-periodic magnetic field and a spatially patterned field created by the substrate. The frequency of the time-periodic field is sufficiently slow that the particles and their assemblies remain always in local energy minima. Prior work by the Authors has established how time-periodic fields of constant magnitude can direct the transport of particles or particle clusters on spatially periodic patterns. Using spatially varying “metamorphic” patterns, the Authors have shown how otherwise identical particles can move in different directions when subject to a common time-periodic field. In the present work, the Authors show how these prior capabilities can be leveraged to control the “synthesis” of particle assemblies of a prescribed size. Through careful selection of the metamorphic pattern and the time-periodic field, they demonstrate that smaller assemblies migrate towards a line on the surface while larger assemblies migrate away from that line. In this way, single particles and smaller assemblies accumulate and assemble to form larger assemblies that migrate away from the “polymerization line”. The design of the time-periodic field is guided by topological arguments that build on previous works by the Authors.

The novel concept of this paper is the field-driven assembly of linear colloidal chains of prescribed length that can be varied predictably by tuning the driving field. Similar structures can be assembled by other means----for example, in a rotating field where the frequency controls the balance of viscous and dipolar forces and thereby the length of the largest stable chain. Nevertheless, the present approach is certainly very clever and sophisticated.

The physical understanding required to demonstrate this capability (namely, understanding and control of “topologically protected colloidal transport”) was developed in series of previous papers by the Authors. Without reading those papers in detail, this Reviewer found the theoretical discussions presented here to be largely inaccessible; however, I have no reason to doubt their validity. As a synthetic strategy for directed colloid assembly, the paper would likely be more impactful if it could be made more accessible to a broader audience.

Some specific questions to consider:

1. As a method for colloidal assembly (beyond a proof-of-concept demonstration), how effective is the proposed approach? The examples provided in Figures 2 and 3 show how the synthesis is supposed to work. How does this synthesis fail when it does not work? Presumably, assemblies

larger than the target may also form. Can the Authors comment on the rate and yield of the colloidal synthesis.

2. What is the period of the time-periodic field? How slow does it need to be to be “adiabatic”?

3. What is the role of Brownian motion? The theoretical discussion of particle transport suggests that particle motion is deterministic and reversible; however, the videos show noisy particle trajectories that appear to follow different paths from cycle to cycle.

Reviewer #3 (Remarks to the Author):

The work presents an intriguing approach to assembling magnetic colloids into chains of adjustable size, introducing a new strategy. The introduction is well-structured and appropriately referenced. The results are adequately described, although there are some details I will address later. However, I have reservations regarding the paper's appeal to the general readership of Nature Communications. Additionally, the theoretical section, particularly the initial two paragraphs of the single particle response subsection, is challenging to comprehend. I suggest the authors attempt to simplify this section further, if feasible.

In summary, this work shows promise for publication in Nature Communications. Nonetheless, the authors should resolve the following concerns before proceeding:

- In some videos, I did not observe the purported transport of monomers (e.g., monomers on the left in 466133_0_video_8290969_s3jqd.mp4 and 466133_0_video_8290968_s3fhq.mp4). This raises doubts about the reliability of the process.
- In Figure 1a, the distinction between juvenile and outgrown bipeds is based on whether they've reached the outgrown length. However, both the blue and magenta aggregates appear to be the same size in this figure.

- In Figure 1b, the subloops 3, 4, and 9 (12, 13, and 16, respectively) of the LNE and LSE loops are labeled differently. The first appears to be divided into two sections (3 and 4), while the second is described with only an index. Is this accurate? Additionally, the LCW loop is not visible in the same figure, and the n-axis is missing.
- What is the typical frequency applied? How does frequency affect the observed phenomena? What is the strength of the applied field?
- How is the transition in substrate A constructed? Can you control the size and shape of transition zone S6?
- What happens if you increase the density regime of particles? Is the phenomenon still observable?
- Is there a limit to the size of chains that the technique allows for adjustment?
- Page 6, column 2, second paragraph: "The walking direction... is perpendicular to the encircled point...". To my knowledge, a direction cannot be perpendicular to a point.

Reviewer #4 (Remarks to the Author):

The work presents a compelling concept using both metamorphic patterns on the bottom substrate and the periodic repetition of specific loops to orient an applied external

Field. This strategy effectively assembles magnetic beads into colloidal chains of desired lengths.

Overall, I find this paper interesting to read. The idea is novel. However, certain aspects require further elucidation before it could meet the standards of publication in Nature Communication.

(1) The terminology "polymerization/polymerize," "synthesis," and "polymerization site" may lead to confusion. While I understand that the authors are trying to draw an analogy between polymerization and their work, it is essential to note that the bipeds are not chemically bound, nor is

there a demonstrated chemical synthesis Therefore, I suggest using “assembly” unless the authors can further demonstrate that those chains are chemically linked.

(2) The absence of a thorough discussion on polydispersity, a common phenomenon in polymers, is notable. The impact of particle concentration on assembly, particularly polydispersity, merits exploration. For example, even if the authors use a specific exit loop, say, $L_{\text{exit},7}$, could the chains always be grown until they reach 7? Only very low particle concentration is used here. It would be better to perform experiments at much higher particle concentrations.

(3) Key questions remain unanswered regarding the width of the polymerization zone, its influence on biped assembly, and the yield of this method. Additional discussions addressing these inquiries would enhance the comprehensiveness of the paper.

(4) Bipeds with chain lengths of 3, 5, and 7 have been illustrated and discussed. What about those chain lengths with even numbers? Also, the authors need to discuss the limits of this method. For example, it appears that the selection between chains of length of n and $n+1$ would become more challenging as n increases. So, what is the comfortable range of chains that can be easily grown and differentiated, and what is not?

(5) A summarizing paragraph either at the end of the experimental sections or the beginning of the theoretical section would provide clarity on the physical and conceptual framework of this work. Given the complexity of the mathematical exposition in the theoretical section, a physical explanation of the mechanism behind juvenile chain growth toward the polymerization site and the departure of overgrown chains would enhance readability and accessibility for a broader audience.

We thank all three referees for their detailed and helpful reviews of our work

Reviewer 2 (Remarks to the Author):

This paper investigates the migration and assembly of paramagnetic colloids moving above a planar substrate due both to a homogeneous, time-periodic magnetic field and a spatially patterned field created by the substrate. The frequency of the time-periodic field is sufficiently slow that the particles and their assemblies remain always in local energy minima. Prior work by the Authors has established how time-periodic fields of constant magnitude can direct the transport of particles or particle clusters on spatially periodic patterns. Using spatially varying “metamorphic” patterns, the Authors have shown how otherwise identical particles can move in different directions when subject to a common time-periodic field. In the present work, the Authors show how these prior capabilities can be leveraged to control the “synthesis” of particle assemblies of a prescribed size. Though careful selection of the metamorphic pattern and the time-periodic field, they demonstrate that smaller assemblies migrate towards a line on the surface while larger assemblies migrate away from that line. In this way, single particles and smaller assemblies accumulate and assemble to form larger assemblies that migrate away from the “polymerization line”. The design of the time-periodic field is guided by topological arguments that build on previous works by the Authors.

The novel concept of this paper is the field-driven assembly of linear colloidal chains of prescribed length that can be varied predictably by tuning the driving field. Similar structures can be assembled by other means—for example, in a rotating field where the frequency controls the balance of viscous and dipolar forces and thereby the length of the largest stable chain. Nevertheless, the present approach is certainly very clever and sophisticated.

We thank the Referee for the constructive criticism and for the for the positive judgement. We respond to all points raised by the Referee in the following.

The physical understanding required to demonstrate this capability (namely, understanding and control of “topologically protected colloidal transport”) was developed in series of previous papers by the Authors. Without reading those papers in detail, this Reviewer found the theoretical discussions presented here to be largely inaccessible; however, I have no reason to doubt their validity. As a synthetic strategy for directed colloid assembly, the paper would likely be more impactful if it could be made more accessible to a broader audience.

Accessibility: We are aware that many colloidal scientists may not be familiar with the language of topology. The language of topology is boosting the field of electronic transport in semiconductors at the moment. As we show here topology has profound implications on the transport of colloids as well. We have to make a decision here not to repeat our previous papers on the subject, but at the same time provide the reader

with a hint about how the topology works in this system. We find the easiest way of doing this is by demonstrating to the reader that we can compute the fences as well as the bifurcation points respectively lines both in control and in transcription space. The determination of the biped travel direction follows straightforwardly from the fences and bifurcation points using the rotating wheel rule.

In order to step by step lead the reader through the topology of single transport we have revised our old figure 4 and now show both action and control space in a less compact way. First, we explain the location of the fences in control space for various symmetry phases and transfer the knowledge of the location of the bifurcation points in these different control spaces to one control space. We have completely rewritten the section of single colloidal particle transport and hope that the text and the figure make it easier to understand for the reader.

In the revised manuscript we also present with our new figure 5 an illustrative cartoon of the rotating wheel rule. Virtual wheels symmetrically attached to two bifurcation lines in transcription space are activated by loops winding around the axes of the wheels. Wheels with axes around which the loop does not wind remain inactive. The transport direction of a biped of a particular length is then given by the rolling direction $\boldsymbol{\omega} \times \boldsymbol{n}$ where $\boldsymbol{\omega}$ denotes the direction of the axis.

Changes:

- we have changed the section (marked C3 in the manuscript) about single colloidal transport and
- revised the figure 4.
- added figure 5
- prepared the reader for the figures 6 and 7 by explaining the rotating wheel rule for bipeds in detail (see paragraph marked C4 in the manuscript)

We hope these changes will help the reader to follow the major arguments concerning the topological nature of the phenomenon.

Some specific questions to consider:

1. *As a method for colloidal assembly (beyond a proof-of-concept demonstration), how effective is the proposed approach? The examples provided in Figures 2 and 3 show how the synthesis is supposed to work. How does this synthesis fail when it does not work? Presumably, assemblies larger than the target may also form. Can the Authors comment on the rate and yield of the colloidal synthesis.*

The synthesis can fail in two ways. There can be insufficient single colloids being transported to the active zone, in which case the biped remains juvenile and gets stuck inside the active zone, waiting forever for the single colloid that would render it outgrown. Since the exit loop for juvenile bipeds are designed to be topologically trivial the juvenile biped will not exit. Such ever remaining juvenile bipeds are already presented in our experimental movies

We have never encountered bipeds in the active zone being longer than outgrown. If this were about to happen, we have designed the exit loops to be topologically trivial for lengths that are more than outgrown but not very much longer. Our synthesis can fail once the outgrown bipeds leaves the active zone and counter propagates the incoming single colloids. In such case the outgrown bipeds may collide with incoming single colloidal particles leading to unwanted further growth. It is for this reason we have to keep the colloidal concentration of single colloids low outside the active zone. The failure is similar if you want to synthesize a product using an enzyme, but the reaction probability is not low without enzyme. The failure of the synthesis via the collision of properly synthesized bipeds with incoming single colloidal particles can in principle be mended using a more complex metamorphic pattern that transports the outgrown bipeds in a second metamorphic direction not interfering with the influx direction of the single colloids. Our topological loops would be more complex for such a pattern and we want to lay the emphasis of this work on the principles of such method not on solving further issues.

We have added the text marked C5 in the manuscript to the discussion reading:
In fact our device functions like an enzyme in biology. Due to the low concentration of colloids no polymer addition reactions are supposed to occur outside the active zone. Polymer addition reactions are wanted at the active line, not elsewhere. By topologically transporting single colloids into the active zone the colloidal concentration is increased there as to increase the reaction rate. The choice of concentration therefore is critical for the proper functioning of the device. Since single colloids are kept at the same distance of at least one unit cell, while transporting them to the active zone, polymer addition reactions of single colloids never happened outside the active zone in our experiments. However once an outgrown biped leaves the active zone it counter propagates to the single colloids and unwanted addition reactions can elongate the length of bipeds beyond the outgrown length. It is possible to design more complex metamorphic patterns where outgrown bipeds may avoid further collisions upon exit and using more sophisticated driving loops. The current work just shows the potential of the method. It produces outgrown bipeds without polydispersivity. The kinetics is of a Michaelis-Menten type saturating at a concentration when all active zones are busy. This is the case once one section of length b_{n^*} of the active line yields roughly one outgrown biped per n^* periods of the loop. Due to the requirement of adiabatic transport our yield is low, however, our precision is high.

This addresses also the rate, the yield and the monodispersity of our method

2. *What is the period of the time-periodic field? How slow does it need to be to be “adiabatic”?*

We answer this question by adding the text marked C7 to the methods section of our manuscript reading: The adiabatic nature is insured by keeping the Mason number of all bipeds below one $\mathcal{M} \ll 1$. The Mason number of an outgrown biped is $\mathcal{M} = 2\pi\eta n^{*3/2}/\mu_0\chi_{\text{eff}}H_{\text{ext}}H_pT_F$ (with η the shear viscosity of the fluid and χ_{eff} the effective magnetic susceptibility of a colloidal particle, and T_F the period of a fundamental loop.). The effect of non-adiabatic driving has been discussed in [15]. In our experiments with $n^* = 7$ we used a modulation period of $T_F = 25$ s for a fundamental loop. Our modulation loop consists of twelve fundamental loops such that largest period used was $T \approx 5$ min for our largest outgrown biped.

3. *What is the role of Brownian motion? The theoretical discussion of particle transport suggests that particle motion is deterministic and reversible; however, the videos show noisy particle trajectories that appear to follow different paths from cycle to cycle. Brownian motion indeed plays a role for details of the path during one loop. The fluctuations however are much smaller than a lattice constant and therefore cannot change the transport displacement over a period. A second effect of Brownian fluctuations is a broadening of the fences and bifurcation lines. We have discussed the effects of Brownian motion in detail in reference [11]. There are, however, different paths from cycle to cycle also because the pattern is changing along the metamorphic direction: with this metamorphism details differ from cycle to cycle, but the step width and direction after completion of a loop remain the same.*

We thank the referee once more for his helpful review of our work.

Reviewer 3 (Remarks to the Author):

The work presents an intriguing approach to assembling magnetic colloids into chains of adjustable size, introducing a new strategy. The introduction is well-structured and appropriately referenced. The results are adequately described, although there are some details I will address later.

We are happy the referee finds our manuscript an “intriguing approach to assembling magnetic colloids into chains of adjustable size.”

1. *However, I have reservations regarding the paper’s appeal to the general readership of Nature Communications. Additionally, the theoretical section, particularly the initial two paragraphs of the single particle response subsection, is challenging to comprehend. I suggest the authors attempt to simplify this section further, if feasible.*

See our answer with headline **Accessibility** to Referee 2

In summary, this work shows promise for publication in Nature Communications. Nonetheless, the authors should resolve the following concerns before proceeding:

2. In some videos, I did not observe the purported transport of monomers (e.g., monomers on the left in 466133 0 video 8290969 s3jhd.mp4 and 466133 0 video 8290968 s3fhq.mp4. This raises doubts about the reliability of the process.

We create our patterns with a lithographic process. Errors in the lithographic process at the edges between black and white enlarge the white areas at the expense of the black areas. The imbalance of the total white and black areas causes a net magnetization $\int \mathbf{M} d^3x_{\mathcal{A}} \neq 0$ that acts the same way as if there were an additional external field normal to the film. In such a case the effect on an apparent modulation loop is to shift the loop crossing away from the equator and giving the northern loops more strength in the S_6 -region than the southern loops. We try to counter these lithographic errors with a tiny offset of the loop crossing that brings the balance back. One can see a balanced behavior in the synthesis of b_3 , b_4 and b_6 and a less balanced behavior in the synthesis of b_2 , b_5 and b_7 .

3. In Figure 1a, the distinction between juvenile and outgrown bipeds is based on whether they've reached the outgrown length. However, both the blue and magenta aggregates appear to be the same size in this figure.

We have redrawn figure 1 with smaller colloidal spheres such that now one can easily count the spheres in the biped; the magenta biped consists of seven, the blue biped of six single colloidal particles.

4. In Figure 1b, the subloops 3, 4, and 9 (12, 13, and 16, respectively) of the LNE and LSE loops are labeled differently. The first appears to be divided into two sections (3 and 4), while the second is described with only an index. Is this accurate? Additionally, the LCW loop is not visible in the same figure, and the n-axis is missing.

All loops are concatenations of loop segments. The concatenation of our branch free two old figure loop segments 3,4 to the concatenated new loop segment 3 of the new figure does not change the correctness of either figure. To avoid confusion we have relabeled the loop segments such that new segment numbers are given only after passing a branch point. We provide a supplementary movie *adfigure1.mp4* showing the flyby of the camera to both action and control space, in which all subloops of our modulation loops including \mathcal{L}_{CW} are visible. We also include the normal vector \mathbf{n} into the axes of control space. In the caption to figure 1 we note: The loop \mathcal{L}_{CW} is not visible in the figure but is a mirror-image of the loop \mathcal{L}_{CE} mirrored at the isomorphic ι -plane. We show it in the supplementary movie *adfigure1.mp4*, that presents the scheme from different perspectives.

5. What is the typical frequency applied? How does frequency affect the observed phe-

nomena? What is the strength of the applied field?

Please, see our answer to question 2 of Referee 2 and the text marked C7 in the methods section of our manuscript

The magnitude of the external field , $H_{\text{ext}} = 4 \text{ kAm}^{-1}$, is given in the subsection **External field** of our methods section

6. How is the transition in substrate A constructed? Can you control the size and shape of transition zone S6?

The magnetization pattern is mathematically computed using Eqs. (1) and (2) using the lattice constant $a = 7 \mu\text{m}$ and the morphing reciprocal vector $\boldsymbol{\mu} = -\mathbf{q}_0/240$ as described in the text. Other morphing reciprocal vectors $\boldsymbol{\mu}$ would result in broader or narrower active zones. In the experiments we cannot easily change the width of the transition zone since this would require the production of another lithographically printed pattern.

7. What happens if you increase the density regime of particles? Is the phenomenon still observable?

Increasing the density of particles destroys the selectivity for outgrown bipeds. Outgrown bipeds are still produced in the described way, however, once they exit the active zone they are set on a counter propagating course to the incoming single colloidal particles. They eventually collide with single particles to grow beyond their outgrown length.

Please, see also our response to question 1 of Referee 2 and the paragraph marked C5 in the discussion of the manuscript

8. Is there a limit to the size of chains that the technique allows for adjustment?

We have included the paragraph marked C6 in the discussion reading:

Theoretically there is no size limit to the production of bipeds of any length. In practice in the experiments there is a size limit. The robustness of the process relies on loops encircling the fences by keeping a distance to the bifurcation locations. The number of bifurcation locations in transcription space increases with the number of bipeds (all transcription loops to bipeds with $n < n^*$ have to fulfill the distance requirement, compare Fig. 6 and 7 which becomes increasingly challenging. The adiabatic period $T \propto n^{*3/2}$ becomes longer and longer and at some point we would need to reduce the modulus of the morphing reciprocal vectors $\boldsymbol{\mu}$, to avoid a biped having one foot in the S_6 the other in the C_6 region. Moreover the total cross section of a biped grows proportional to its size, such the concentration of single colloids must be reduced to avoid unwanted collisions of the biped after its exit of the active zone.

9. Page 6, column 2, second paragraph: "The walking direction... is perpendicular

to the encircled point...". To my knowledge, a direction cannot be perpendicular to a point.

We ammended the text, what we meant is:

The walking direction for the grown up bipeds for each of the loops \mathcal{L}_{CE} and \mathcal{L}_{CW} is perpendicular to the external field vector pointing to the encircled point of each of the loops

We appreciate the careful report that helped to improve our manuscript.

Reviewer 4 (Remarks to the Author):

The work presents a compelling concept using both metamorphic patterns on the bottom substrate and the periodic repetition of specific loops to orient an applied external Field. This strategy effectively assembles magnetic beads into colloidal chains of desired lengths. Overall, I find this paper interesting to read. The idea is novel.

We thank the referee for the positive judgement of our paper.

However, certain aspects require further elucidation before it could meet the standards of publication in Nature Communication.

(1) The terminology "polymerization/polymerize," "synthesis," and "polymerization site" may lead to confusion. While I understand that the authors are trying to draw an analogy between polymerization and their work, it is essential to note that the bipeds are not chemically bound, nor is there a demonstrated chemical synthesis Therefore, I suggest using "assembly" unless the authors can further demonstrate that those chains are chemically linked.

Yes we try to draw an analogy between polymerization and our work. Of course the bipeds are not chemically bound. Instead of using the word assembly we stick to our originally used wording, however, we state explicitly on page 3, where the polymerization is first introduced in the paragraph marked C1:

The active line serves a polymerization line for monomeric single colloidal particles into polymeric bipeds being rods of several colloidal spheres. The bond between the monomers of the polymerized bipeds is a physical dipolar interaction bond and not a chemical bond.

We hope that this makes it clear that our mesoscopic system is a physical analogue to a chemical process.

(2) The absence of a thorough discussion on polydispersity, a common phenomenon in polymers, is notable. The impact of particle concentration on assembly, particularly polydispersity, merits exploration. For example, even if the authors use a specific exit loop, say, $L_{exit,7}$, could the chains always be grown until they reach 7? Only very low

particle concentration is used here. It would be better to perform experiments at much higher particle concentrations.

We now discuss the polydispersity. Our process produces completely monodisperse bipeds exiting the active zone. Our process should be considered a catalytic respectively enzymatic process. The active zone on our pattern serves as the active site. In order for the process to work, polymerization addition reactions should only occur in the active zone not outside this zone. Unfortunately it would not be better to perform experiments at much higher particle concentrations. We have already shown that not all bipeds in the active zone for the specific exit loop, say, $\mathcal{L}_{\text{exit},7}$ grow to their outgrown size. The specificity of the reaction, however, is that only properly finished bipeds exit the active zone, juvenile bipeds remain in the active zone expecting further single colloids to enter the active zone. Of course this expectation of the juvenile biped may be never met. Increasing the concentration of particles destroys the selectivity for outgrown bipeds. Outgrown bipeds are still produced in the described way, however, once they exit the active zone they are set on a counter propagating course to the incoming single colloidal particles. They eventually collide with single particles to grow beyond their outgrown length. The low concentration of single colloidal particles avoids such unwanted reactions outside the active zone and is essential for the selectivity for the production of monodisperse bipeds.

We have added on page 3 the text marked C1

In analogy to the active site of an enzyme, we call the iso-phase line with locally S_6 -symmetric pattern at $\varphi = 2\pi/12$ the active line (cyan in Fig. 1a). The active line serves as a polymerization line for monomeric single colloidal particles into polymeric bipeds being rods of several colloidal spheres. The bond between the monomers of the polymerized bipeds is a physical dipolar interaction bond and not a chemical bond. The concentration of colloids outside the active region must be small enough to prevent polymerization reactions outside the active zone; polymerization addition reactions shall only occur in the active zone. In the active zone our modulation loop will ensure that the reaction proceeds toward the wanted outgrown biped length and not toward unwanted longer biped lengths. Juvenile bipeds having lengths below the outgrown length shall not leave the active zone before they are outgrown.

and we have added the paragraph marked C5 to the discussion.

Please, see also answer to question 1 of Referee 2

(3) Key questions remain unanswered regarding the width of the polymerization zone, its influence on biped assembly, and the yield of this method. Additional discussions addressing these inquiries would enhance the comprehensiveness of the paper.

The width of the polymerization zone has to be small compared to the length of an outgrown biped. In our pattern we use a metamorphic reciprocal vector of size $\mu = -\mathbf{q}_0/240$ that keeps consecutive S_6 lines far apart without making the active zone too large. As mentioned in the response to the previous question our yield is low but our

precision (monodispersity) is high.

See the changes discussed in the previous question and in question 1 of Referee 2

(4) Bipedes with chain lengths of 3, 5, and 7 have been illustrated and discussed. What about those chain lengths with even numbers? Also, the authors need to discuss the limits of this method. For example, it appears that the selection between chains of length of n and $n+1$ would become more challenging as n increases. So, what is the comfortable range of chains that can be easily grown and differentiated, and what is not?

We include now experiments for even $n^* = 2, 4$, and $n^* = 6$ outgrown biped lengths (see the paragraph marked C2 and new supplementary movies *b2synthesis.mp4*, *b4synthesis.mp4* and *b6synthesis.mp4*)

We also include a paragraph marked C6 dicussing the limits of the method in the discussion.

See also answer to question 8 of Referee 3

(5) A summarizing paragraph either at the end of the experimental sections or the beginning of the theoretical section would provide clarity on the physical and conceptual framework of this work. Given the complexity of the mathematical exposition in the theoretical section, a physical explanation of the mechanism behind juvenile chain growth toward the polymerization site and the departure of overgrown chains would enhance readability and accessibility for a broader audience.

Please, see our answer with headline **Accessibility** to Referee 2

We thank the referee for his careful review of our work and for helping us improve the quality of this manuscript

further minor changes:

- we have renamed polymerization line into active line and polymerization zone into active zone
- We have ammended the text where it describes features of now six instead of three different exit loops and tabled their parameters in table I and table II.

We thank all three referees for their detailed and helpful reviews of our work

REVIEWERS' COMMENTS

Reviewer #2 (Remarks to the Author):

The Authors have made several additions and clarifications in response to the Reviewer comments, which have improved the clarity and accessibility of the manuscript. In particular, the new section on the topological transport of a single particle helps to more clearly explain the design principles used to achieve polymer size control.

While I do not claim to follow the topological arguments in detail, I am sufficiently impressed by the reported capability of polymer size control that I will be studying this and related papers in more detail. There are clearly opportunities for further applications in colloidal assembly and the development of “physically intelligent” microrobots.

The new capability reported here is a significant advance over previous reports of directed transport on similar metamorphic patterns as it demonstrates the tunable sensitivity of the transport speed and direction to the size of the colloidal chains.

I recommend publication of the revised manuscript in Nature Communications.

Reviewer #3 (Remarks to the Author):

After the implemented changes in the manuscript, I believe it now meets the criteria set by Nature Communications for publication.

Reviewer #4 (Remarks to the Author):

I am satisfied with the revision, therefore, recommending its publication in Nat Comm.

We thank all three referees for their detailed and helpful reviews and for their support of our work